EMBO
Molecular Medicine

# Ultra-sensitive molecular residual disease detection through whole genome sequencing with single-read error correction

Xinxing Li [1,12], Tao Liu [2,12], Antonella Bacchiocchi [3,12], Mengxing Li [4,12], Wen Cheng [4], Tobias Wittkop [5], Fernando L Mendez [5], Yingyu Wang [5], Paul Tang [5], Qianqian Yao [6], Marcus W Bosenberg [3,7,8], Mario Sznol [7,9], Qin Yan [7,8,10], Malek Faham [5], Li Weng [5✉], Ruth Halaban [3,7✉], Hai Jin [4✉] & Zhiqian Hu [1,11✉]

## Abstract

While whole genome sequencing (WGS) of cell-free DNA (cfDNA) holds enormous promise for detection of molecular residual disease (MRD), its performance is limited by WGS error rate. Here we introduce AccuScan, an efficient cfDNA WGS technology that enables genome-wide error correction at single read-level, achieving an error rate of $4.2 \times 10^{-7}$, which is about two orders of magnitude lower than a read-centric de-noising method. The application of AccuScan to MRD demonstrated analytical sensitivity down to $10^{-6}$ circulating variant allele frequency at 99% sample-level specificity. AccuScan showed 90% landmark sensitivity (within 6 weeks after surgery) and 100% specificity for predicting relapse in colorectal cancer. It also showed 67% sensitivity and 100% specificity in esophageal cancer using samples collected within one week after surgery. When AccuScan was applied to monitor immunotherapy in melanoma patients, the circulating tumor DNA (ctDNA) levels and dynamic profiles were consistent with clinical outcomes. Overall, AccuScan provides a highly accurate WGS solution for MRD detection, empowering ctDNA detection at parts per million range without requiring high sample input or personalized reagents.

**Keywords** Circulating Tumor DNA; Molecular Residual Disease; Single-read Error Correction; White Blood Cell-free; Whole Genome Sequencing
**Subject Categories** Cancer; Chromatin, Transcription & Genomics; Methods & Resources

## Introduction

Molecular residual disease (MRD) refers to the small amount of cancer cells persisting after treatment (Ignatiadis and Reinholz, 2011). Timely and sensitive measurement of MRD is critical for recurrence risk assessment and treatment decisions (Gale et al, 2022). Circulating tumor DNA (ctDNA) has emerged as a promising real-time biomarker for MRD detection and monitoring. Studies have shown that the levels of cancer-specific somatic mutations in ctDNA correlate with tumor stage, burden, and response to therapy across tumor types (Li and Liang, 2020; Lipson et al, 2014). ctDNA provides a more sensitive and specific measure of MRD compared to other blood-based cancer biomarkers, such as circulating tumor cells and cancer antigens (Honore et al, 2021; Pantel and Alix-Panabieres, 2017).

There are currently two main strategies for ctDNA-based MRD detection: (1) the tumor-naive approach, which tests MRD samples for changes known to be enriched in cancers such as methylation and common somatic mutations (Parikh et al, 2021); (2) the tumor-informed approach, which requires a tumor sample to identify patient-specific variants and then tests MRD samples for those variants (McDonald et al, 2019; Semenkovich et al, 2023).

The tumor-naive approach uses a universal panel to test plasma samples for a cancer signal without the need to acquire and sequence the tumor (Cristiano et al, 2019; Jamshidi et al, 2022; Parikh et al, 2021). While these tests are convenient, they tend to have moderate sensitivity. For instance, a methylation-based cancer detection test demonstrated 50% analytical sensitivity with a circulating tumor allele fraction of $3.1 \times 10^{-4}$ at 98% specificity (Jamshidi et al, 2022). Furthermore, an analysis combining both methylation and mutation signals showed a clinical sensitivity of 55.6% in predicting recurrence in colorectal cancer (CRC) using

[1]Department of Gastrointestinal Surgery, Tongji Hospital, Tongji University School of Medicine, Shanghai 200065, P. R. China. [2]Department of Thoracic Surgery, Peking University First Hospital, Beijing 100034, China. [3]Department of Dermatology, Yale University School of Medicine, New Haven, CT, USA. [4]Department of Thoracic Surgery, Shanghai Changhai Hospital, Second Military Medical University, Shanghai 200433, China. [5]Department of Research and Development, AccuraGen Inc, San Jose, CA 95134, USA. [6]Department of Medical Science, Shanghai YunSheng Medical Laboratory Co., Ltd, Shanghai 200437, China. [7]Yale Cancer Center, Yale School of Medicine, New Haven, CT, USA. [8]Yale Center for Immuno-Oncology, Yale School of Medicine, New Haven, CT, USA. [9]Department of Internal Medicine, Section of Medical Oncology, Yale University School of Medicine, New Haven, CT, USA. [10]Department of Pathology, Yale University, New Haven, CT, USA. [11]Department of General Surgery, Changzheng Hospital Naval Medical University, Shanghai 200003, P. R. China. [12]These authors contributed equally as first authors: Xinxing Li, Tao Liu, Antonella Bacchiocchi, Mengxing Li. ✉E-mail: lweng@accuragen.com; ruth.halaban@yale.edu; jinhai@smmu.edu.cn; 2105203@tongji.edu.cn

 

plasma collected at landmark time point (4 weeks after surgery) (Parikh et al, 2021).

The tumor-informed approach utilizes tumor-specific somatic mutations from a patient's tumor for MRD analysis, which is highly specific and sensitive. Factors that impact its sensitivity include the accuracy of somatic mutation calls in the tumor and plasma samples, and the total number of cell-free DNA (cfDNA) molecules interrogated, which is the product of the number of somatic variants tracked and the average molecular depth obtained through sequencing.

Tumor-informed approaches can either use bespoke or off-the-shelf MRD tests. A bespoke MRD assay is designed after tumor sequence data is available and follows a limited number of variants through ultra-deep sequencing (Bratman et al, 2020; Zhao et al, 2023). The sequencing of a bespoke panel can be exhaustive; therefore, the average molecular depth is primarily constrained by the amount of available input material (Abbosh et al, 2017; Coombes et al, 2019; Kotani et al, 2023; Zhao et al, 2023). For example, Signatera™, a tumor-informed NGS-based multiplex PCR assay that tracks 16 personalized markers, claimed an analytical sensitivity of 81.3–96.1% at the limit of detection (LOD) of $10^{-4}$ when up to 66 ng of DNA was used (Coombes et al, 2019). Tumor-informed personalized MRD assays that target large numbers of markers and apply error correction using unique molecular identifiers (UMI) or duplex sequencing have shown LODs below $10^{-4}$ (Cohen et al, 2021; Moding et al, 2020; Zhao et al, 2023). PhasED-Seq uses multiple somatic mutations in individual DNA fragments to lower the background noise to less than $10^{-6}$ and claimed LOD down to the parts per million (PPM) level given enough phased variants (Kurtz et al, 2021). While the tumor-informed bespoke MRD approach can achieve very high sensitivity, the requirement of a personalized design substantially increases turnaround time (TAT) and creates considerable logistical challenges.

The tumor-informed off-the-shelf method uses the same assay for all patients. Without the need for patient-specific reagents, it shares the low TAT of a tumor-naive approach and offers much simpler logistics than the bespoke method. The challenge is generating an off-the-shelf assay that covers enough of the genome at a low enough error rate (Chaudhuri et al, 2017; Chen et al, 2021; Li et al, 2022; Qiu et al, 2021). Pre-designed MRD panels targeting cancer-related genes typically use UMI with deep sequencing to achieve high accuracy in variant calling, but the number of markers that these panels track for each patient are very limited (Chaudhuri et al, 2017; Chen et al, 2021; Qiu et al, 2021). For example, a 130 kb panel covering 139 critical lung cancer-related genes only captures a median of 2 mutations per patient (range: 1–8 mutations) (Qiu et al, 2021).

Whole genome sequencing (WGS) assays have recently emerged as an innovative approach for cancer screening (Bruhm et al, 2023; Cristiano et al, 2019; Zhang et al, 2022) and MRD detection (Hallermayr et al, 2022; Nordentoft et al, 2024; Tan et al, 2022). Tumor-informed WGS MRD assays use genome breadth to supplement sequencing depth for sensitivity, overcoming the limitation of input sample amount. However, the standard UMI error correction, which relies on having multiple reads per input molecule, would be cost prohibitive on a WGS scale (Illumina, 2023; Kinde et al, 2011; Wang et al, 2019). Zviran et al used a read-centric Support Vector Machine model to reduce the error rate for WGS somatic single-nucleotide variants (SNV) to $4.96 \times 10^{-5}$ (Zviran et al, 2020). By capitalizing on the cumulative signal of thousands of somatic mutations observed in the tumor genome, they reported longitudinal sensitivity of 91% at 92% specificity in bladder cancer using accumulated ctDNA status (Nordentoft et al, 2024). Other WGS technologies using duplex sequencing have demonstrated an ultra-low error rate at $<10^{-7}$ level, however, these methods suffer from low conversion rates, making a low LOD difficult to achieve (Alexandre Pellan et al, 2022; Bae et al, 2023; Liu et al, 2024). There is a need for an efficient and cost-effective genome-wide error correction method to enable WGS for MRD detection with high sensitivity and specificity.

DNA concatemers generated via rolling circle amplification (RCA) of cfDNA physically link replicated copies, allowing error correction at single read level (Wang et al, 2020; Xu et al, 2017). The combination of RCA with repeat confirmation eliminates both PCR and sequencing errors. Concatemer sequencing has demonstrated a high conversion rate and sensitivity for liquid biopsy applications, including therapy selection and cancer screening, when adapted for targeted sequencing through hybrid-capture or amplicon workflows (Wang et al, 2023; Wang et al, 2020). In this study, we present AccuScan, a WGS solution for ctDNA detection that utilizes concatemer sequencing for genome-wide single-read error suppression, enabling fast and sensitive MRD detection and monitoring in cancer patient plasma samples.

## Results

### AccuScan's genome-wide error suppression enables detection of ultra-low ctDNA levels

The AccuScan assay workflow (Fig. 1A) is optimized for low input cfDNA, efficiently capturing double-stranded, single-stranded and nicked DNA (Wang et al, 2023). cfDNA is denatured and circularized through ligation, followed by whole-genome amplification using RCA, which generates concatemer molecules containing tandem copies of the original template. These concatemer products are sequenced using paired-end 150 (PE150) base sequencing and aligned to the human reference genome. Sequences of each copy within a read pair are compared. A change from the reference that is consistent in all copies and supported by at least two copies is a presumed variant; an inconsistent change is likely due to a PCR or sequencing error and is removed. To assess the efficiency of error correction by AccuScan, we sequenced the cfDNA samples from healthy donors ($n = 3$) using both regular WGS and AccuScan WGS (Fig. 1B). The observed average error rate was $9.4 \times 10^{-4}$ for regular WGS without filtering; the error rate was reduced to $2.8 \times 10^{-5}$ when requiring overlap and concordance between read pairs. AccuScan with concatemer error correction had an average error rate of $4.2 \times 10^{-7}$, which is ~2000-fold lower than the unfiltered WGS data, and ~67-fold reduction when compared to the read-pair corrected WGS (Fig. 1B; Appendix Fig. S1). The AccuScan error rate can be further reduced to $3.1 \times 10^{-7}$ when sequenced using the single-end 300 (SE300) base sequencing (Fig. EV1).

We performed simulations to understand the impact of error rate on ctDNA detection under different circulating variant allele frequencies (cVAF), sequencing depths, and numbers of markers (Figs. 2 and EV2). A statistical model that calculates the probability

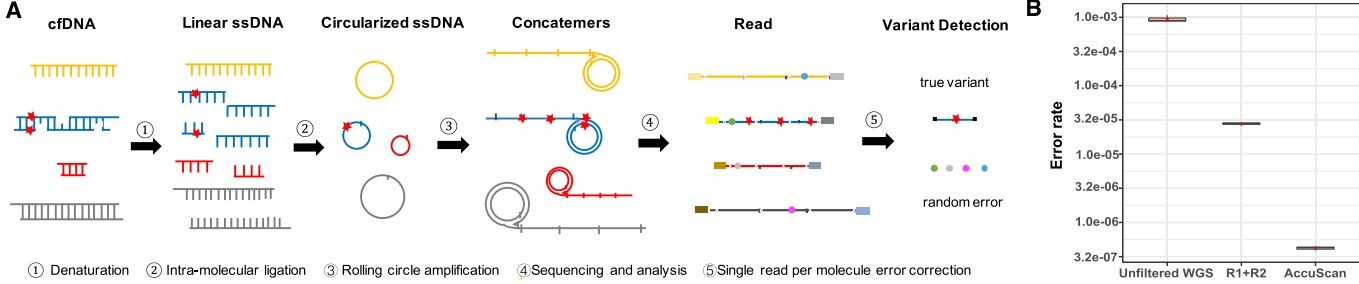

**Figure 1. AccuScan: WGS through concatemer error correction.**

(A) AccuScan workflow. Cell-free DNA (cfDNA) is denatured and circularized, followed by rolling circle amplification using random primers to create concatemers. These concatemer products are sequenced and aligned to the human reference genome. A non-reference base that is consistent across all repeats will be called as a mutation. Random errors that are inconsistent between repeats are removed. Examples of circularized molecules are shown for a subset of cfDNA fragments. (B) Error rate of WGS on healthy human cfDNA samples ($n = 3$) measured by unfiltered reads, read1 and read2 corrected reads and AccuScan. The line in the middle of the boxplot represents the median value; the box borders reflect the interquartile range (25th to 75th percentiles); and the whiskers indicate 1.5 times IQR; the red diamonds indicate the mean value. ssDNA, single-stranded DNA; WGS, whole genome sequencing; R1 + R2, variants called by requiring read1 and read2 overlap and concordance.

of observing expected variants at specific marker loci is used to predict the presence of ctDNA. Sensitivity was calculated as the number of positives predicted over the total number of simulations for each condition, under the nominal specificity setting of 99%. Figure 2A shows the sensitivity from simulations using 10K markers with either $2.8 \times 10^{-5}$ or $4.2 \times 10^{-7}$ error rates under sequencing depths ranging from 10× to 100×. Either decreasing the error rate or increasing the sequencing depth improved the detection rate. For example, with an error rate of $4.2 \times 10^{-7}$ and a 20× sequencing depth, there is a 96% detection rate at a cVAF of $2.5 \times 10^{-5}$, but when the error rate is $2.8 \times 10^{-5}$, 100× sequencing depth is required to achieve a similar detection rate at the same cVAF. The specificity remains above 98.8% across all conditions when cVAF is set to 0, consistent with the nominal specificity setting of 99% in the model (Fig. 2B).

We then measured the analytical performance of AccuScan using healthy sample mixtures. cfDNA from three different healthy "test" donors was independently titrated into cfDNA from a separate healthy "background" donor at seven different concentrations. This process created mixed samples containing germline SNVs from the "test" samples, with cVAF ranging from $1 \times 10^{-4}$ to $1 \times 10^{-6}$. These 21 cfDNA mix samples were then sequenced to 60× using AccuScan with 10 ng input DNA per reaction and we assessed our ability to detect the germline SNVs of the "test" donor from background. In each dilution, out of the over 100K SNVs distinguishing the test and background samples (Table EV1), we performed the test using randomly selected subsets of 2K, 5K, 10K, or 20K SNVs as markers, chosen to represent the variant-type profile typical of CRC tumors (Appendix Fig. S2, Methods). Marker selection was repeated 1000 times per condition and MRD testing was run with 99% nominal specificity. The observed specificities were ≥99% for 2K, 5K, 10K, or 20K marker conditions (Fig. 2C). The observed sensitivity at a level of ≥$2.5 \times 10^{-5}$ cVAF was greater than 99% for all conditions tested. At 10 PPM, corresponding to $1 \times 10^{-5}$ cVAF, the average detection rate was 38% for 2K markers, 77% for 5K markers, 96% for 10K markers, and 100% for 20K markers (Fig. 2D; Appendix Fig. S3).

The analytical sensitivity of AccuScan was further confirmed by mixing cfDNA from a melanoma patient with cfDNA from a healthy donor (Fig. EV3A). The original cancer cfDNA sample had

a cVAF of 1.1% as measured by droplet digital PCR (ddPCR) of a BRAF V600E mutation present in the primary tumor (Fig. EV3B). Dilutions were made for 5 different expected frequencies, ranging from $1 \times 10^{-3}$ to $2 \times 10^{-6}$. ddPCR of BRAF V600E was performed to confirm the $1 \times 10^{-3}$ dilution (Fig. EV3C). The diluted cancer samples were sequenced by AccuScan with 10 ng input per reaction. We used the start and end positions of a cfDNA molecule as the unique identifier to count the number of unique molecules (Methods). The number of tumor variant molecules detected and the observed cVAFs correlate with the expected variant frequencies in the cfDNA samples (Fig. 2E; Appendix Fig. S4). The observed detection rate is 100% for samples with cVAF of $1 \times 10^{-3}$, $1 \times 10^{-4}$ and $1 \times 10^{-5}$, 67% (2/3) at cVAF of $5 \times 10^{-6}$ and 33% (1/3) at cVAF of $2 \times 10^{-6}$ (Fig. 2E). AccuScan sequencing of a negative control (cfDNA from a healthy donor) yielded negative results in both replicates.

To further corroborate the specificity of AccuScan, we utilized over 1.3 M tumor variants from tumor tissue samples and simulated datasets of variants to be tested in plasma samples from mismatched individuals. The MRD model and its nominal specificity of 99% was fixed before this test and did not rely on empirical data (Methods). This analysis served the purpose of measuring the specificity of the MRD model in plasma samples from cancer patients. The specificity analysis aimed to assess all plasma samples using the same total number of cfDNA fragments, calculated as the average depth multiplied by the number of variants. Given that not all cfDNA samples were sequenced to the exact same depth, we normalized the number of selected variants to a depth of 60× in our sampling. From the pool of ~1.3 million variants, we randomly sampled subsets equivalent to 2K, 5K, 10K, or 20K variants at a depth of 60× to test MRD calls in mismatched patient plasma samples. For example, if a plasma sample has an average depth of 30×, sampling 10K variants would be necessary to match an equivalent variant count of 5K at a depth of 60×. We performed 2000 random samplings of mismatched variants for each combination of equivalent variant count level and plasma sample. The specificity for each plasma sample at each equivalent variant count level is calculated as the number of negative MRD calls out of the total of 2000 random samplings (Methods). The observed average specificities are 99.6%, 99.3%, 99.1%, and 98.9%,

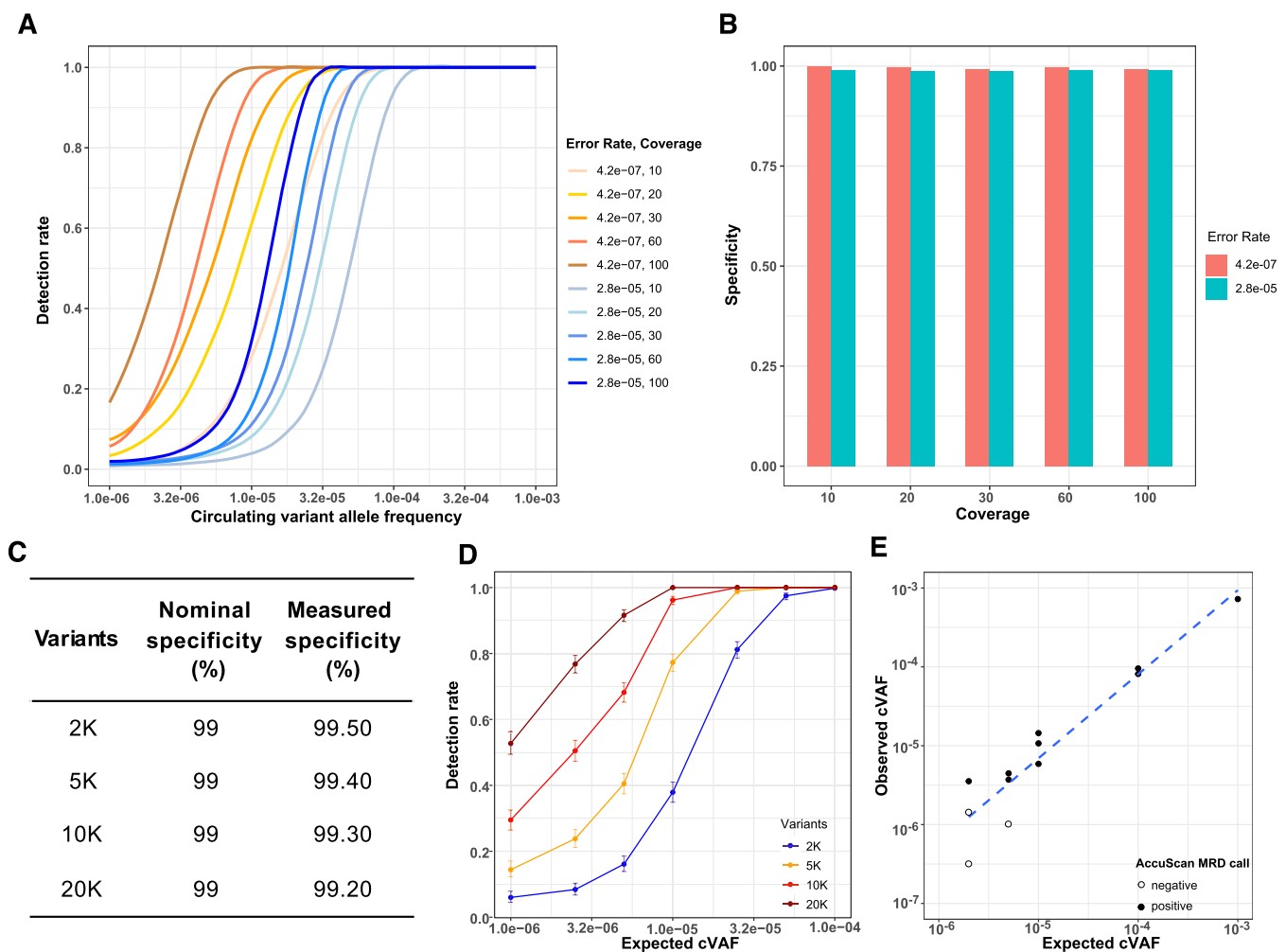

**Figure 2. Analytical sensitivity and specificity of AccuScan.**

(A) Simulation using 10K markers and two error rates to predict the theoretical detection rates under different sequencing coverages as a function of circulating variant allele frequency (cVAF). Detection rate is calculated as the fraction of tests that are called molecular residual disease (MRD) positive with the nominal specificity set at 99%. (B) Simulation using 10K markers and two error rates to predict the theoretical specificity with the nominal specificity setting at 99%. Specificity is calculated as the fraction of tests that are called MRD negative when cVAF is 0. (C, D) Observed analytical specificity (C) and sensitivity (D) with serial dilutions of healthy cell-free DNA (cfDNA) mixtures simulating cVAF from $1 \times 10^{-4}$ to $1 \times 10^{-6}$. cfDNA from healthy donors ($n = 3$) was titrated independently into cfDNA from a different healthy "background" donor. Samples were processed by AccuScan with 10 ng input, and tested for MRD by sampling 2K, 5K, 10K, or 20K single nucleotide polymorphisms, 1000 times each, as tumor-specific markers. The error bars in (D) represent the likelihood-ratio-based 95% confidence intervals of the observed sensitivity. (E) Observed cVAF in serial dilutions of a melanoma cfDNA sample with expected cVAF from $1 \times 10^{-3}$ to $2 \times 10^{-6}$. Experiments were performed with one test at $1 \times 10^{-3}$, two replicates at $1 \times 10^{-4}$, three replicates at $1 \times 10^{-5}$, $5 \times 10^{-6}$, and $2 \times 10^{-6}$ cVAFs.

for 2K, 5K, 10K, and 20K equivalent variant count levels, respectively. These results suggest that the AccuScan assay and analysis achieve the intended performance for patient plasma samples using tumor variants (Appendix Fig. S5).

## Identification of tumor-specific variants using a white blood cell-free workflow

A tumor-informed MRD test uses tumor-specific variants as markers for tracking the disease. The most commonly used strategy for tumor-specific variant identification is through paired sequencing of the tumor tissue DNA and a matched normal DNA sample, for example, DNA from white blood cells (WBC) of the patient (Cancer Genome Atlas Research, 2011; Saunders et al, 2012). By

comparing the sequences of matched tumor and normal DNA, paired sequencing can effectively remove germline variants, which can significantly outnumber somatic variants. However, this method requires additional sample processing and sequencing (Fig. 3A). To simplify the MRD workflow, we explore the feasibility of omitting WBC sequencing and using the sequencing data from post-treatment plasma samples to pair with tumor tissue sequencing for germline variant removal and somatic variant identification (Fig. 3B).

The challenge with this WBC-free approach is to distinguish between germline variants and potential cancer variants in post-treatment plasma samples. We addressed this issue by assuming that (1) When the tumor fraction is low, germline variants and somatic variants will be represented by different numbers of

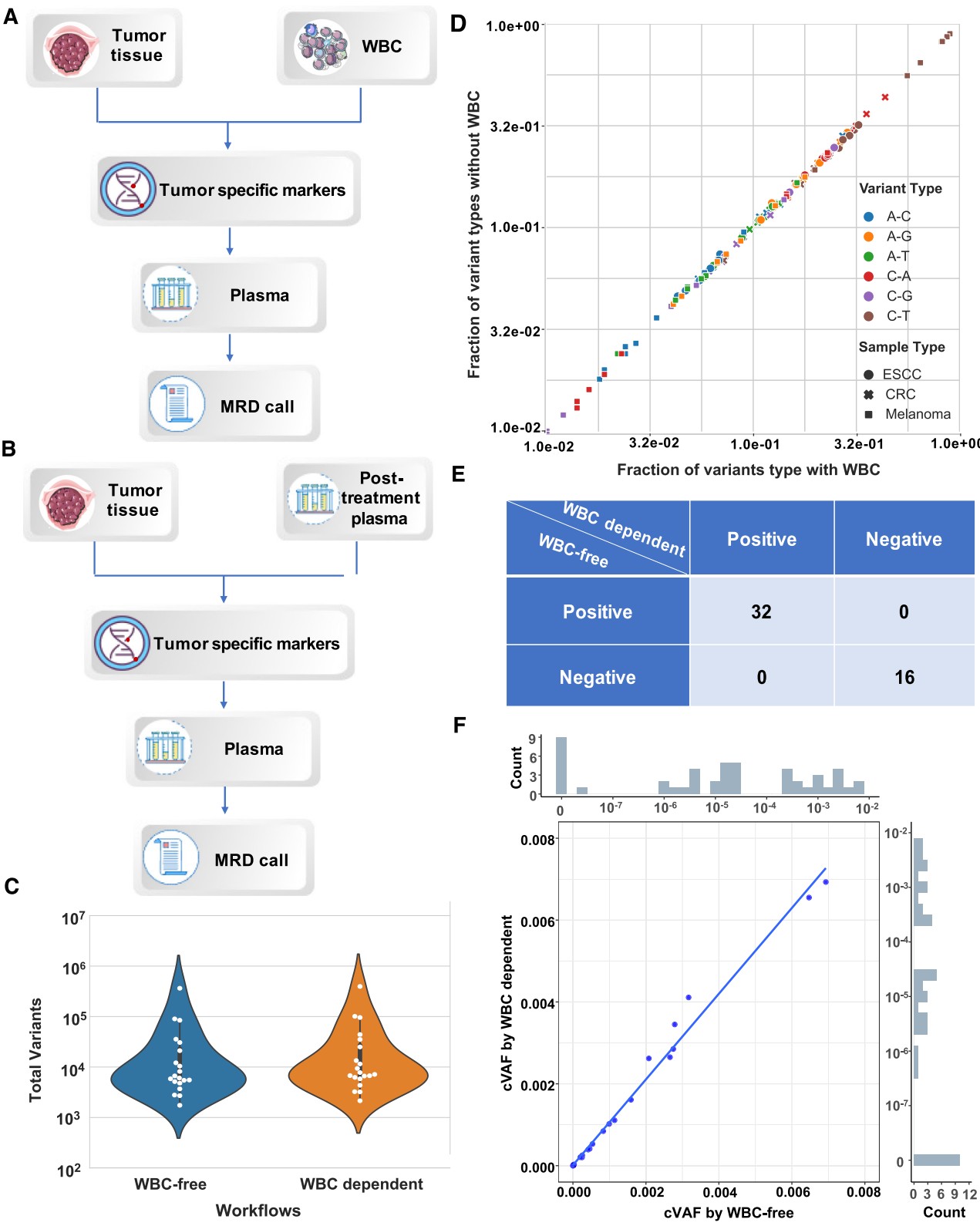

**Figure 3. WBC-dependent and WBC-free workflows for AccuScan MRD detection.**

(A) The white blood cell (WBC) dependent workflow uses whole genome sequencing (WGS) data from WBC to remove germline and clonal hematopoiesis of indeterminate potential (CHIP) variants found in the tumor tissue, keeping tumor-specific markers for molecular residual disease (MRD) test of the plasma samples. (B) The WBC-free workflow uses WGS data from the low-tumor burden post-treatment plasma samples to remove germline and high frequency CHIP variants from the tumor tissue. Any tumor variants that are found in the plasma samples with more than one cell-free DNA molecules are removed from the tumor variant list. The remaining variants from the tumor tissue WGS result are used as tumor-specific markers for MRD test in plasma samples. (C) Total number of tumor-specific markers identified by WBC-free ($n = 20$) and WBC-dependent ($n = 20$) sequencing. Each dot represents one patient; the line in the middle of the boxplot represents the median value; the box borders reflect the interquartile range (IQR, 25th to 75th percentiles); and the whiskers indicate 1.5 times IQR. (D) Comparison of the variant profile for tumor-specific markers identified using two workflows. (E, F) Comparison of the AccuScan MRD call (E) and the observed circulating variant allele frequency (cVAF) (F) in the same plasma samples using tumor-specific markers identified by two workflows. Panel (E) listed the number of MRD positive and MRD negative calls by WBC-dependent and WBC-free workflows. 48 plasma samples from 6 colorectal cancer (CRC) patients, 6 esophageal squamous cell carcinoma (ESCC) patients and 8 melanoma patients were tested. Source data are available online for this figure.

molecules in the post-treatment plasma samples. We modeled the distribution of the expected number of variant molecules observed under different cVAFs using a Poisson model (Appendix Fig. S6). With 40–60× sequencing of cfDNA extracted from a low-tumor-burden (cVAF <0.1%) plasma sample, germline SNVs (cVAF = 50% or 100%) are typically observed at ≥2 unique cfDNA molecule levels, while tumor-specific variants (cVAF <0.1%) are generally found at <2 unique cfDNA molecule levels (Appendix Fig. S6). Therefore, we may remove variants found with 2 or more molecules in the post-treatment plasmas from the tumor tissue sequencing results to obtain the list of tumor-specific mutations. (2) In plasma samples with high tumor burden, a significant fraction of the tumor variants might be observed at ≥2 unique cfDNA molecule levels. However, as shown by Poisson model, at cVAF = 15%, we will still be able to observe multiple variants with one molecule (Appendix Fig. S6), suggesting that the effect on MRD sensitivity is likely to be minimal. The primary impact would be an underestimate of the tumor variant count and the observed cVAF in plasma. (3) At very high cVAF (cVAF > 15%), the Poisson model predicts that the majority of cancer variants will be observed at the level of ≥2 unique cfDNA molecules. Consequently, comparison of the tumor tissue sequencing and plasma AccuScan data will remove most of the variants found in tissue, resulting in few or no tumor-specific variants as MRD markers. Given that tumor tissue samples typically harbor thousands of cancer mutations, the observation of minimal or no tumor variants at the single molecule level would suggest a high tumor fraction in the plasma. This hypothesis can be confirmed by other markers for MRD detection, such as copy number variation or cfDNA fragment size, which are well suited for cancer detection at high tumor fraction.

To test the feasibility of this approach, we compared the performance of a tumor-WBC workflow with a WBC-free workflow using matched tumor tissue, WBC and plasma samples collected from 20 cancer patients (Table EV2, Fig. 3A,B). The number of tumor-specific variants and variant type profiles found by the two different workflows are shown in Fig. 3C,D. Overall, the number of mutations identified by both methods was very similar, as was the variant-type profile of the mutations identified. AccuScan MRD analysis of the plasma samples ($n = 48$, including 18 pre-treatment samples and 30 post-treatment samples) from these 20 patients showed identical MRD calls under either workflow (Fig. 3E, Table EV2), and the cVAF values were strongly correlated ($R^2 = 0.99$, Fig. 3F). These results suggest that post-treatment plasma can be used instead of WBC for the identification of tumor-specific variants.

In addition to germline variants, another potential complication in identifying tumor-specific variants is the presence of clonal hematopoiesis of indeterminate potential (CHIP) variants. CHIP variants are somatic mutations present in hematopoietic cells. It has been shown that the variant allele frequencies (VAF) of CHIP mutations are strongly correlated in the matched cfDNA and WBC samples (Razavi et al, 2019). At 60× sequencing depth, the rule of removing variants with ≥2 unique cfDNA molecules will filter CHIP variants if their cVAF is >3% (2/60) in plasma. However, this rule will not remove CHIP variants with cVAFs below 3% in plasma. Meanwhile, most CHIP mutations are not observed in tumor tissue with high VAF (>5%). The probability of having a CHIP mutation found in the tumor tissue with high enough VAF (>5%) and simultaneously in cfDNA with one unique molecule is small. To prevent false positive calls caused by CHIP mutations, which can be mistaken as tumor-specific mutations present in post-treatment plasma under rare circumstances, sequencing the matching WBC sample when a plasma sample is marginally called MRD positive may be beneficial.

## MRD detection and prognostic value in post-surgical patients

We next evaluated the performance of AccuScan for MRD detection in post-surgical gastrointestinal cancer patients, including 32 patients with CRC and 17 patients with esophageal squamous cell carcinoma (ESCC).

The CRC patients were at diverse clinical stages (22% stage I, 38% stage II, 34% stage III, 6% stage IV) and received radical surgery (Table EV3). Formalin-fixed paraffin-embedded (FFPE) tissues were available from all patients. For the 15 patients with available WBC, we used the tumor-WBC workflow to identify tumor-specific variants; for all other patients, we used the first post-operative (post-OP) plasma samples for the WBC-free workflow (Fig. 3B; Table EV2). We identified a median of 5820 tumor-specific variants per patient in FFPE tissues (2148—265,800, Fig. 4A), which corresponds to ~2 mutations/Mb.

Of the 32 patients, 26 had plasma samples collected before surgery and 28 had plasma samples collected at landmark (within six weeks after surgery). There were 7 additional plasma samples collected at time points later than six weeks after surgery (Table EV3). The median follow-up time in this CRC cohort was 24.13 months [interquartile range (IQR): 18.5–36]. ctDNA was detected in all the 26 pre-operative (pre-OP) cfDNA samples with a median cVAF of $5.2 \times 10^{-4}$ (IQR: $6.6 \times 10^{-5}$–$2.4 \times 10^{-3}$) (Fig. 4B). 34.4% (11/32) of patients had ctDNA detected in the post-OP

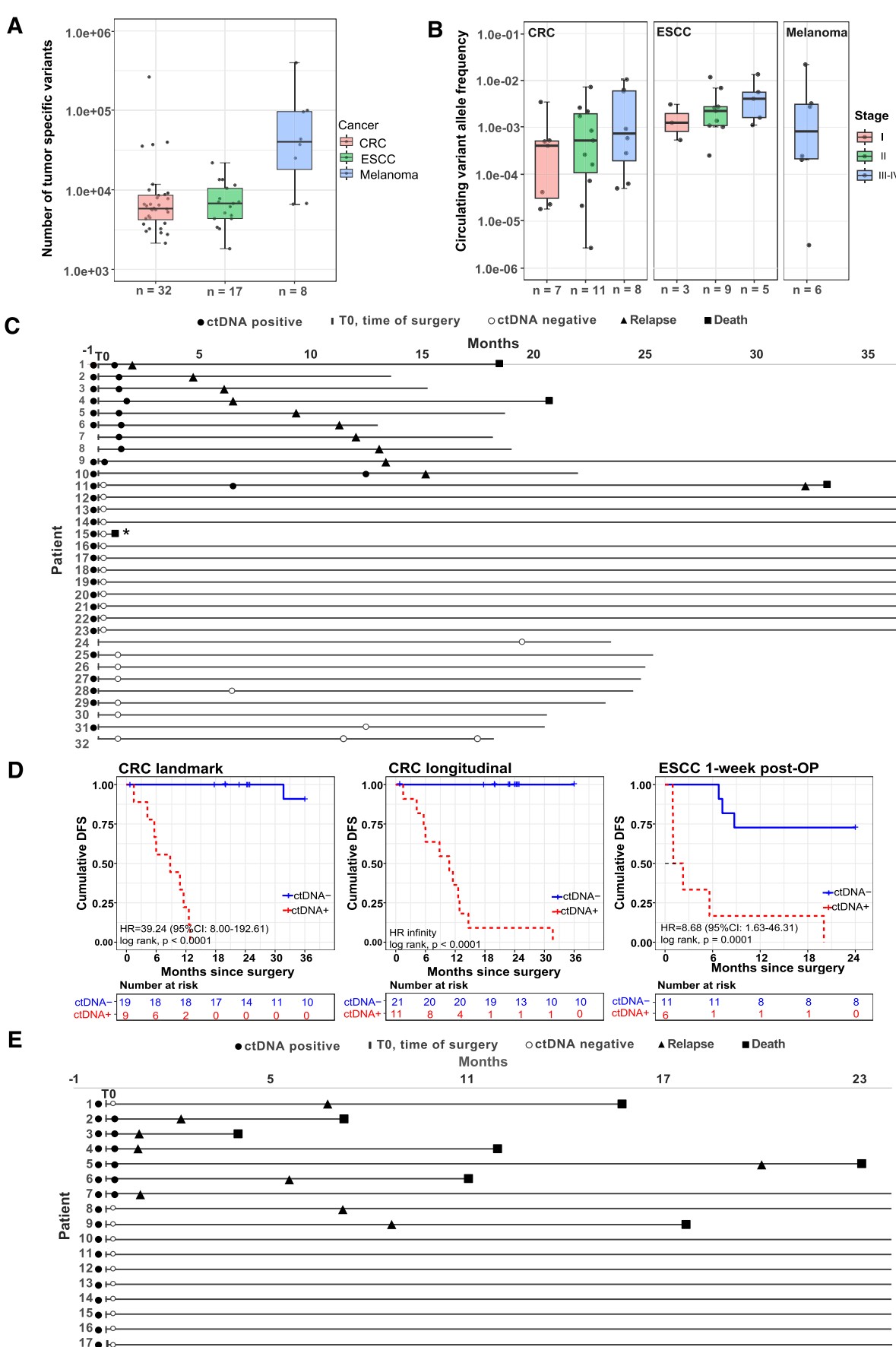

◀ **Figure 4. AccuScan molecular residual disease detection in clinical samples.**

(A) Number of tumor-specific variants identified in colorectal cancer (CRC, $n = 32$), esophageal squamous cell carcinoma (ESCC, $n = 17$) and melanoma patients ($n = 8$).
(B) Observed circulating variant allele frequency (cVAF) in pre-treatment plasma samples from CRC ($n = 26$), ESCC ($n = 17$) and melanoma patients ($n = 6$).
(C) Swimmer plot of 32 CRC patients who undergo radical surgery. *The patient died due to non-cancer related reasons. (D) Kaplan–Meier disease-free survival (DFS) analysis for CRC surgical patients using blood samples collected at landmark time point (within 6 weeks after surgery) only (left) or longitudinal samples collected overtime (middle), and for ESCC surgical patients using blood samples collected within 1 week after surgery (right). Patients who are circulating tumor DNA (ctDNA) positive in the post-operative (post-OP) plasma samples showed significantly shorter DFS. (E) Swimmer plot of 17 ESCC patients. All patients received curative surgery. Patient blood samples were collected before surgery and one week after surgery. ctDNA+, ctDNA positive; ctDNA-, ctDNA negative; HR, hazard ratio; CI, confidence interval. Data information: For (A, B), each dot represents one patient, the line in the middle of the boxplot represents the median value; the box borders reflect the interquartile range (IQR, 25th to 75th percentiles); and the whiskers indicate 1.5 times IQR, with the lower line representing 25th percentile − 1.5*IQR and the upper line representing 75th percentile + 1.5*IQR. Source data are available online for this figure.

samples (Table EV3). All patients that were ctDNA positive in the post-OP samples relapsed within 3 years after surgery (Fig. 4C,D). The post-OP ctDNA positive patients had shorter disease-free survival (DFS) than ctDNA negative patients [hazard ratio (HR), 39.24, 95% confidence interval (CI): 8.00–192.61; log-rank $p < 0.0001$] (Fig. 4D). The median DFS of the ctDNA positive patient group was 10.8 months (IQR: 5.8–12.7), with 63.64% (7/11) of ctDNA positive patients had a recurrence within one year, and 90.91% (10/11) of ctDNA positive patients relapsed within two years. One of the ctDNA-positive patients, patient #11, was ctDNA negative at the first landmark timepoint, converted to ctDNA positive at 6 months post-OP and then relapsed at 32 months. Patients that were ctDNA negative at all post-OP time points were progression-free during the follow-up period (up to 36 months) (Fig. 4C). Taken together, these results suggest 90% (95% CI: 55.5–99.8%) sensitivity, 100% (95% CI: 81.5–100%) specificity and 96.3% (95% CI: 81.0–99.9%) accuracy at landmark (within 6 weeks after surgery), 100% (95% CI: 71.5–100%) sensitivity, 100% (95% CI: 88.1–100%) specificity, and 100% (95% CI: 91.2–100%) accuracy with longitudinal monitoring for predicting CRC recurrence.

The ESCC cohort included patients from stages I–III (18% stage I, 53% stage II, 29% stage III) who received curative-intent surgery (Table EV4). FFPE, WBC, pre-OP plasma, and post-OP (within 1 week after surgery) plasma samples were collected from all patients. Using the paired tumor and WBC samples, we identified a median of 6768 tumor-specific variants per patient (Fig. 4A). The follow-up time of this ESCC cohort ranged from 4.03 months to 24 months. ctDNA was detected in all 17 of pre-OP samples with a median cVAF of 0.27% (IQR: 0.13–0.55%) (Fig. 4B). In the post-OP plasma samples, ctDNA was detected in 35.29% (6/17) of the patients, with a median cVAF of $1.3 \times 10^{-4}$ (IQR: $1.9 \times 10^{-5}$–$1.1 \times 10^{-2}$). ctDNA positive patients had shorter DFS than ctDNA negative patients (HR, 8.68, 95% CI: 1.63–46.31; log-rank $p = 0.0001$) (Fig. 4D). All the patients with ctDNA positive post-OP samples (6/6, 100%) had a disease recurrence within 2 years of surgery; 5 of 6 (83%) patients had a recurrence within one year (Fig. 4E). In contrast, of the 11 patients with ctDNA negative post-OP sample, 8 patients remained disease-free and were followed for 24 months, and 3 patients relapsed. ctDNA detection at 1-week post-OP had 66.7% (95% CI: 29.9–92.5%) sensitivity, 100% (95% CI: 63.1–100%) specificity and 82.4% (95% CI: 56.6–96.3%) accuracy in predicting ESCC recurrence.

In summary, ctDNA detection using AccuScan is highly sensitive and specific in identifying MRD and correlates well with tumor recurrence in both CRC and ESCC patients.

## ctDNA monitoring during immunotherapy

Advances in immune checkpoint blockade (ICB) have significantly improved the survival of patients with advanced melanoma (Carlino et al, 2021). However, only about half of the patients respond to ICB (Wolchok et al, 2022). There is an urgent need for methods of prognosis and monitoring for patients undergoing immunotherapy. We explored the potential use of AccuScan for immunotherapy monitoring in a pilot study with advanced melanoma. A total of 22 plasma samples were collected from 8 melanoma patients, including 6 samples collected before any treatment (Table EV5). WGS of the paired tumor and WBC DNA samples identified a median of 40,580 SNVs, with an average of 88,906 tumor-specific SNVs per patient in tissue samples (Fig. 4A). AccuScan analysis of the pre-treatment plasma samples showed that all six pre-treatment samples were ctDNA positive, with a minimum cVAF measured as low as $2.82 \times 10^{-6}$ (Fig. 4B, Table EV6).

Of the 8 melanoma patients, 6 had at least two plasma samples collected during ICB treatment. We therefore measured ctDNA kinetics and compared the results with radiological data in these 6 patients. For patients 1 through 5, radiographic changes matched the cVAF changes measured by AccuScan (Fig. 5, Table EV6). Patient 1 showed rapid decline of cVAF from baseline to C2, and clearance of ctDNA at C4 time point. Patient 2 was ctDNA positive before surgery, converted to ctDNA negative after surgery and maintained ctDNA negative by C3 of adjuvant ICB therapy. Both patients had sustained complete response with no disease recurrence through the monitoring period. Patients 3 and 4 had very high cVAF levels in all plasma samples. For Patient 3, cVAF increased from 12.5% at baseline to 16% before C2 and computed tomography (CT) scan detected disease progression two months later. Patient 4 observed tumor regression by imaging at 2.6 months after C1, while AccuScan test suggested cVAF increased to ~0.7% at 4-month timepoint, and 2.5 months later, CT scan confirmed tumor progression. Patient 5 was ctDNA positive with persistently low cVAF (~$1 \times 10^{-5}$) in plasma samples taken before and during adjuvant ICB treatment, which is consistent with CT scans showing stable lung nodules.

Patient 6 did not have a pre-treatment sample. The first plasma sample was collected after C4 of adjuvant therapy with nivolumab, followed by 6 samples collected during ipilimumab/nivolumab therapy (Fig. 5A). Based on the MRD model, the first sampling timepoint was ctDNA negative with cVAF below $3.8 \times 10^{-6}$ (Table EV6), while CT scan detected a 4 mm lung nodule. The cVAF elevated to $1.4 \times 10^{-5}$ at the second blood-draw (Table EV6),

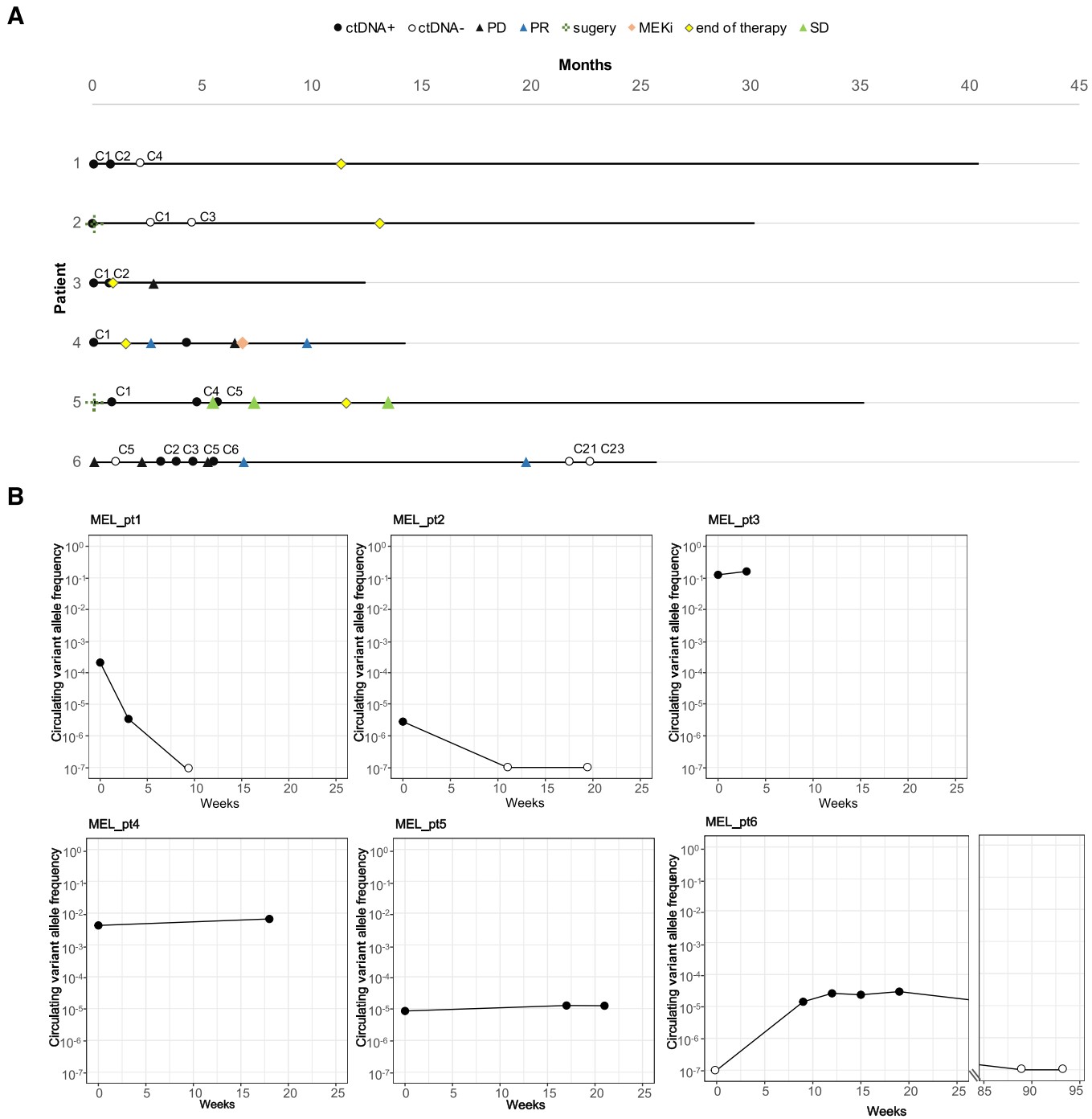

**Figure 5. AccuScan for treatment monitoring of melanoma patients.**

(A) Swimmer plot of six melanoma patients who undergo surgery and immunotherapy. (B) Dynamic change of circulating tumor DNA (ctDNA) levels in melanoma patients over time. PD progressive disease, PR partial response, SD stable disease, MEKi MEK inhibitor, C cycle, e.g., C2 for cycle 2. Source data are available online for this figure.

which was consistent with the increased tumor burden measured by the second CT scan. ctDNA level stabilized after the third blood test, while the imaging data showed continuous tumor progression. The fourth scan, which was taken 1.5 months after the fifth ctDNA test, showed excellent partial response, and the patient reached near

complete response and ctDNA clearance 13 months later (Fig. 5B). These results indicate that the observed progression by third imaging was likely to be pseudo-progression, and the early stabilization of ctDNA level may signal that patient was responding to immunotherapy.

In summary, a total of 117 plasma samples from 57 cancer patients were processed in this study. Of all the cancer plasma samples analyzed in this study, 96.6% (113/117) samples had a total cfDNA amount <60 ng, and 17.1% (20/117) of samples had a total cfDNA amount <10 ng (Table EV7, Fig. EV4). Of the cancer plasma samples that are tested ctDNA positive, about 30% are with cVAF $<1 \times 10^{-4}$, and ~9% are below 10 PPM (Fig. EV5).

## Discussion

This is the first report of applying genome-wide concatemer error correction to tumor-informed MRD detection and monitoring in cancer patients. AccuScan combines RCA and linked reads to remove artifacts introduced during library preparation and sequencing, demonstrating highly efficient and accurate measurement of ctDNA in both contrived samples and clinical samples.

Scarcity of ctDNA remains as the major challenge for MRD detection. As most of the cfDNA molecules are from WBCs, the fraction of tumor-derived DNA fragments can be significantly lower than 0.01% in the post-treatment patient plasma samples (Pantel and Alix-Panabieres, 2019). Detecting tumor-specific DNA at such low frequency requires techniques with high sensitivity and specificity. Tumor-naive MRD tests using epigenetic signals showed moderate sensitivity at cVAF $>1 \times 10^{-4}$ level. The sensitivity of ctDNA detection can be significantly improved with tumor-informed MRD assays, by either sequencing all the molecules exhaustively at selected genomic loci or tracking large number of tumor-specific mutation markers with relatively shallow depth. The first approach often requires personalized reagents, which increases logistical challenges and TAT. In addition, it requires high input DNA amount, such as >60 ng, which may not be always available in clinical settings (Coombes et al, 2019). With the drop of sequencing price, WGS has emerged as an attractive solution for MRD given its relatively simple workflow and low sample input requirement. A WGS approach allows tumor and plasma samples to be sequenced in parallel, enabling fast turnaround. The amount of DNA a WGS test requires can be 10 ng or lower. When combined with a home-blood collection device, it may greatly simplify logistics and improve patient experience.

The sensitivity of a WGS-based MRD test is constrained by the WGS error rate. Simulation data showed that a WGS test with a reduced error rate can achieve a lower LOD; and for the same LOD, a test with fewer errors requires significantly fewer reads compared to a WGS test with a higher error rate (Fig. EV2). AccuScan reduces the SNV error rate to less than 1 in 2 million with PE150 sequencing, enabling LOD with 95% probability (LOD95) of detecting tumor-specific mutations at 50 PPM with 10× coverage using 10K SNV markers, and detection of tumor-specific mutations down to ≤10 PPM at 60× coverage, while maintaining a high sample-level specificity of 99%. Comparable analytical sensitivity was achieved with contrived samples, including mixtures of normal cfDNA and dilutions of tumor cfDNA into normal cfDNA. The AccuScan error rate can be further improved with longer sequencing reads. Under PE150 sequencing, the R1 and R2 sequences of a short library molecule with only one copy of the cfDNA can be mistakenly treated as two independent repeats for variant confirmation. This may lead to some of AccuScan's residual errors. Preliminary data showed that SE300 sequencing

further lowered the error rate (Fig. EV1), suggesting that longer reads can effectively clean up residual errors by ensuring correction with 2 or more copies of independent repeats. It is worth noting that the error rates shown in Fig. 1B were measured using plasma samples from three healthy individuals, aiming to compare regular WGS method with AccuScan. The error rate observed in a cancer patient sample may be higher than those in the healthy plasma samples. Therefore, it is important to calculate the error rate for each cancer patient sample when applying AccuScan to MRD detection to accurately assess noise at the individual level.

The sequencing depth needed for a given sensitivity requirement is inversely correlated with the number of mutations for a WGS-based MRD assay. The average mutation rate, which meters the average WGS depth needed for MRD, is ~4.0 mutations/Mb measured across cancer types (Alexandrov et al, 2020; Chalmers et al, 2017; Lawrence et al, 2013). Relatively low mutation rate was observed in pediatric malignancies (median 1.7 mutations/Mb), while diseases associated with mutagen exposure such as lung cancer or melanoma showed high mutation rate (median mutation rate 7.2 mutations/Mb and 13.5 mutations/Mb, respectively) (Chalmers et al, 2017). In our study, we observed the average mutation rates of ~2.6 mutations/Mb, 5.6 mutations/Mb and 26.7 mutations/Mb for ESCC, CRC and melanoma respectively. Given such mutation rates, AccuScan can reach 10 PPM or lower with an average of 60× sequencing coverage. In clinical practice, AccuScan sequencing depth can be adjusted based on cancer types and mutation rate of the patient to achieve an optimal cost-benefit balance while accommodating different application needs.

The sensitivity and specificity requirements for an MRD test depend on the specific applications. For example, the National Comprehensive Cancer Network guideline recommends starting adjuvant chemotherapy for high-risk CRC patients no later than 6–8 weeks after surgery (Benson et al, 2021). The landmark timepoint, which is before week six after surgery, is key for clinicians to decide between escalation or de-escalation of therapy. Current commercial offerings showed an average sensitivity of 41–56% at landmark for CRC (Parikh et al, 2021; Reinert et al, 2019; Tarazona et al, 2019). Faster TAT and higher sensitivity at landmark are needed to give oncologists the time and confidence to de-escalate treatment while ensuring most, if not all, high-risk patients receive needed adjuvant therapy following their surgery. On the other hand, during longitudinal monitoring for patients who have shown partial response or complete response, TAT is not as critical as the landmark time point, but high specificity is crucial for avoiding anxiety, unnecessary exposure to toxic side effects and financial burden associated with the treatments. In this report, AccuScan demonstrated analytical sensitivity at PPM level with contrived samples and achieved a landmark sensitivity of 90% for predicting relapse after surgery with 100% specificity in retrospective CRC patient samples. Taken together, these results indicate that AccuScan test has the potential to achieve very high sensitivity and specificity for MRD detection. It is important to note that the data presented here are from proof-of-principle studies. Large, blinded clinical studies with consecutively collected patient cohorts and sufficient follow-up (i.e., ≥3 years) are needed to confirm and validate the clinical performance of AccuScan.

WGS data offers not only SNV results, but also other rich molecular information, including copy number variants, structural variants, tumor mutation burden (TMB), mutation signatures, as well as epigenetic

information such as fragmentomics. A WGS technology with a low error rate like AccuScan would significantly improve the efficiency and accuracy for TMB and tumor mutation signature detection in cfDNA. In addition, studies have shown that using machine learning to integrate genetic information from multiple somatic variant types significantly improved the performance of tumor-informed MRD detection (Zviran et al, 2020); and the combination of genetic and epigenetic information, such as SNV and fragmentomics signals, may improve the sensitivity of cancer early detection (Bruhm et al, 2023; Wang et al, 2023). AccuScan as a single-strand sequencing technology captures small cfDNA fragments representing regulatory protein binding footprints (Wang et al, 2023). Combining the mutation and fragmentomics signal from AccuScan data holds great potential for developing a tumor-naive test for MRD detection and cancer screening.

Overall, the AccuScan MRD test demonstrated a high success rate in processing cfDNA samples across a wide range of input DNA amounts, with the lowest sample input of less than 5 ng (Table EV7, Fig. EV4). We observed high sample-level specificity of 99%, with 100% sensitivity in all pre-treatment samples from CRC, ESCC, and melanoma. We observed a landmark sensitivity of 90% for CRC post-surgical patients, and a sensitivity of 67% for ESCC using plasma samples collected within one week after surgery. The current performance of AccuScan is based on a simple statistical model using SNV information alone. There is potential for further enhancement through baseline modeling with the accumulation of cfDNA WGS data and the incorporation of machine learning strategies.

# Methods

**Reagents and tools table**

| Reagent/Resource | Reference or Source | Identifier or Catalog Number |
| --- | --- | --- |
| **Chemicals, Enzymes and other reagents** | | |
| *Single strand DNA ligase* | Xu et al, 2017 | *Supplementary Material and Methods* |
| **Software** | | |
| *The R Foundation for Statistical Computing v4.2.2* | *The Comprehensive R Archive Network* | https://www.r-project.org/ |
| *Python v3.9* | *Python Software Foundation* | https://www.python.org/ |
| *BWA mem* | | https://bio-bwa.sourceforge.net/bwa.shtml, Li, 2013 |
| Sentieon® TNseq® | | https://www.sentieon.com/products/ |
| Nextflow | | https://www.nextflow.io/, Di Tommaso et al, 2017 |
| **Other** | | |
| *KAPA Hyper Prep Kit* | *Roche* | *Cat #KK8504* |
| *NEBNext® Ultra™ II Ligation Module* | *New England Biolabs* | *Cat #E7595L* |
| *NEBNext® Ultra™ II End Repair/dA-Tailing Module* | *New England Biolabs* | *Cat #E7546L* |
| *DNBSEQ-T7* | *MGI* | https://en.mgi-tech.com/products/instruments_info/5/ |
| M220 ultrasonicator | Covaris | *Cat #500295* |

| Reagent/Resource | Reference or Source | Identifier or Catalog Number |
| --- | --- | --- |
| Qubit fluorometer 3.0 | Thermo Fisher Scientific | *Cat #*Q33216 |
| 2100 bioanalyzer | Agilent | *Cat #G2939BA* |
| TIANamp Genomic DNA Kit | TIANGEN | *Cat # DP304-03* |
| High Pure FFPET DNA Isolation Kit | Roche | *Cat #6650767001* |
| Qiagen DNeasy Blood and Tissue Kit | QIAGEN | *Cat # 69506* |

## Experimental design

Participants were recruited at four hospitals. The CRC patients' samples were collected from Shanghai Changzheng Hospital (IRB: CZ2017-251) and Tongji Hospital (IRB: K-KYSB-2021-005) from July 2017 to February 2023. The ESCC patients' samples were collected from Shanghai Changhai Hospital (IRB: CHEC2020-021) from June 2020 to December 2021. The melanoma patients' samples were collected from Yale School of Medicine (IRB: 0609001869) from March 2019 to February 2023. A total of 58 subjects including 32 CRC patients, 17 ESCC patients and 9 melanoma patients were included in this report. Inclusion and exclusion criteria were listed in Table EV8. This study was conducted according to the principles set out in the WMA Declaration of Helsinki and the Department of Health and Human Services Belmont Report. It was approved by the Institutional Review Boards and written informed consent was obtained prior to the initiation of the study. Blood samples were collected in blood collection tubes (BD Vacutainer K2EDTA tubes for CRC and ESCC, catalog no. 367525; BD sodium heparin tubes for melanoma patients, catalog no. 367874). FFPE tumor tissues were collected from resected CRC and ESCC biopsies. Fresh frozen tumor tissues were collected from melanoma tumor biopsies. The outcomes of the CRC and ESCC patients were not blinded, while the outcomes of the melanoma patients were blinded to the analysts when doing the ctDNA and bioinformatics analysis.

## Plasma DNA processing

Blood samples were centrifuged within 4 h after blood drawing. For ESCC and CRC, plasma samples were separated by two steps: (1) 10 min at $1900 \times g$ at 4 °C, WBCs were collected from the middle phase and stored at −80 °C; (2) the upper phase of the first centrifugation was collected and centrifuged for another 10 min at $16,000 \times g$ at 4 °C. cfDNA extraction procedures were in accordance with the manufacturer's instructions. Isolation of cfDNA from 1 to 2 mL of plasma was performed using MiniMax High Efficiency Cell-Free Isolation Kit (catalog no. A17622CN-384, Apostle) and eluted in 80 μL Tris-EDTA buffer. The extracted cfDNA was qualified by Qubit fluorometer 3.0 (catalog no. Q33216, Thermo Fisher Scientific) and 2100 bioanalyzer (catalog no. G2939BA, Agilent).

For melanoma patient samples, plasma was collected after spinning the tubes at $800 \times g$ for 10 min. The collected plasma is then spun again at $450 \times g$ for 10 min before being aliquoted and

stored at –80 °C. PBMC are isolated using lymphoprep (catalog no. 07801_c, STEMCELL Technology).

## Tissue genomic DNA (gDNA) processing

For CRC and ESCC patients, the TIANamp Genomic DNA Kit (catalog no. DP304-03, TIANGEN) was used to extract gDNA of WBCs and frozen tissue. gDNA was extracted from paraffin sections of tumor samples by High Pure FFPET DNA Isolation Kit (catalog no. 6650767001, Roche). The melanoma tumor and matching WBC DNA were extracted from fresh frozen samples with Qiagen DNeasy Blood and Tissue Kit (catalog no. 69506, QIAGEN).

## Library preparation and sequencing

The gDNA of WBCs and tissue samples was sonicated into short fragments with a peak around 300 bp by M220 ultrasonicator (catalog no. 500295, Covaris), 100 ng fragmented gDNA was used for library construction using KAPA Hyper Prep Kit (catalog no. KK8504, Roche) following the manufacturer's protocol. The gDNA libraries were sequenced on DNBSEQ-T7 (MGI) with paired-end 150 bp mode.

cfDNA library preparation was performed as described previously (Wang et al, 2023; Wang et al, 2020; Xu et al, 2017). Briefly, cfDNA molecules were heat denatured to become single-stranded and then circularized directly through intra-molecular ligation using a single-strand DNA (ssDNA) ligase, which is effective in circularizing DNA molecules ≥15 bases in size (Xu et al, 2017). The circularized cfDNA molecules are then amplified by whole genome rolling circle amplification (RCA) using random hexamers as primers. The whole genome RCA products were sonicated into fragments with a peak at around 500 bp by ultrasonicator, followed by library construction using NEBNext® Ultra™ II Ligation Module (catalog no. E7595L, NEB) and NEBNext® Ultra™ II End Repair/dA-Tailing Module (catalog no. E7546L, NEB). The cfDNA libraries were sequenced on DNBSEQ-T7 (MGI) with DNBSEQ-T7RS High-throughput Sequencing Set (FCL PE150).

## Tissue variant pipeline and tissue-informed marker selection algorithm

WGS reads were demultiplexed using Complete Genomics sequencer software and processed by Sentieon® TNseq® for variant calling. The tumor-normal mode was used when WBC data was available; otherwise, the tumor-only mode was used. When WBC data was not available, we removed potential germline and high allele frequency CHIP variants by pairing the tumor sequencing data with sequencing data from the matching post-treatment plasma sample, which is expected to have a very low tumor burden. Tumor variants detected with more than one unique molecule in the post-treatment plasma sample are filtered out. C to T substitutions at CpG sites were removed from the variant list for MRD analysis.

## AccuScan pipeline, error correction algorithm, and error rate measurement

Variant calls using concatemers have been previously described (Xu et al, 2017). In summary, paired-end fastq files were processed

using a nextflow pipeline (Di Tommaso et al, 2017) and Amazon Web Services (AWS) cloud computing to obtain variant calls and molecule depth estimates at each position, as well as an error rate estimate for the given plasma sample. First, reads were aligned to the reference genome (hg38) using bwa mem (Li, 2013). Molecule boundaries are determined from neighboring alignments within the same read-pair and can be used for quantification. Molecule depth is calculated at each position and requires repeat confirmation for variant and wild-type calls.

Each position within the molecule boundaries is evaluated for repeat confirmation, where only positions in the read with high read quality (Q-score > 24) are considered. Base calls that were consistently supported within a read-pair by all tandem copies and supported by at least two copies were considered repeat confirmed. Differences from the reference calls that are repeat confirmed are called as true variants; inconsistent differences between tandem copies were considered as random errors and discarded. Repeat confirmed reference calls are counted towards molecule depth at the genomic location together with repeat confirmed variant calls.

For computational reasons, error rates were calculated for a set of randomly selected ~200 Mb positions of the genome. To ensure that this doesn't bias the estimate, we performed tests 20 times using different sets of 200 Mb positions. In this test, the overall error rate ranges from $4.0 \times 10^{-7}$ to $4.5 \times 10^{-7}$ with an average of $4.3 \times 10^{-7}$. This variation is small and would have minimal impact on our MRD calls (Appendix Fig. S1). Error rates were calculated for every possible SNV in the selected loci, except for C to T variants at CpG sites, which were excluded for overall error rate calculation. To remove most germline variants in the error estimate of plasma samples, we filter variants using the gnomad (3.1) database (Karczewski et al, 2020). In addition, variants that would not be considered for MRD due to being in variant positions where two or more variant molecules were observed, are also excluded in this calculation. The remaining variants are sorted into variant types and error rates are calculated for each variant type as the sum of all observed variant molecules divided by the total sum of molecules interrogated for each variant type, i.e., only variants that pass our previous filter criteria are contributing towards the enumerator and denominator in this equation.

## AccuScan MRD algorithm

The AccuScan MRD algorithm is applied on a plasma sample with error corrected variant calls. It further requires a list of target SNVs and evaluates whether the observed number of variants in the plasma overlapping this list of markers exceeds the expectation from a background error model. The error model is measured for each plasma as described above. For each target variant position, the algorithm collects the repeat confirmed molecule depth and distinguishes between reference and variant counts. Unless the number of detected variants is exceedingly high, thus indicating a high tumor burden sample, one can assume that many or most of the detected somatic variants are expected to occur only with a single molecule (Appendix Fig. S6), therefore variants with more than one cfDNA molecule detected in the plasma are excluded for the MRD call in a low tumor burden sample. In addition to a positive or negative MRD call, the relative

amount of tumor DNA is estimated using a probabilistic model, where the number of tumor variant molecules is modeled as a random variable that depends on the sequencing depth, error rates, and cVAF.

We estimate the cVAF by maximizing the likelihood with respect to the variant allele frequency (*VAF*), with the constraint of (*VAF* ≥ 0) and the observed error rate of the sample. For each observation of a variant site *i*, there is a probability $p_i$ of observation of the variant, which we can approximate as $p_i \sim VAF + e_i$, where *VAF* is the variant allele frequency and $e_i$ is the error rate for that site and variant. We can combine all variant sites of the same variant type with the error rate $e_v$ and compute the number of variant observations as a binomial random variable for that variant type with binomial parameter $p_v$, $p_v \sim VAF + e_v$. We assume that the variant types v∈V are determined by the type of mutation as follows:

$$V = \{'C-T','C-G','C-A','A-T','A-G','A-C'\}$$

the log-likelihood function for combining independent binomial distributions of all different variant types is

$$K + \sum_{v \in V} \left( c_v \cdot \ln p_v + (d_v - c_v) \cdot \ln(1 - p_v) \right),$$

$c_v$ here *v* is the number of observations of variant type, $d_v$ is the total depth for that type of variant site, and *K* does not depend on $p_v$ or *VAF*. We obtain the point estimate for cVAF by maximizing the log-likelihood: we find the value of VAF at which the derivative of the log-likelihood with respect to VAF is 0. When this value is positive, it is our estimate; otherwise, the point estimate is 0.

$$\sum_{v \in V} \frac{c_v \cdot (1 - p_v) - (d_v - c_v) \cdot p_v}{p_v \cdot (1 - p_v)}$$

Our null hypothesis is that variant observations are strictly due to errors. To test this hypothesis, we first consider that the null hypothesis corresponds to an extreme of the range of possible values for the parameter (*VAF* ≥ 0 *always, and for the null hypothesis H0 : VAF* = 0), then find the difference in log-likelihoods at the point estimate of *VAF* and at 0, and reject the null hypothesis when the cumulative distribution of a $\chi^2$ distribution with 1 degree of freedom evaluated at twice the difference in log-likelihoods is greater than 0.98. This corresponds to a *P*-value < 0.01 for the mixture distribution (delta and $\chi^2$) that corresponds to testing as null an extreme of the possible values of the parameter (VAF = 0), and therefore a specificity of 0.99 (99%) for the likelihood ratio test (Chernoff, 1954). 95% CIs were obtained in the standard way based on the likelihood ratio: The maximum likelihood over values of VAF is obtained and its logarithm is computed. The non-negative values of VAF such that twice the difference of log-likelihood with the maximum is less than 3.841 are included in the confidence bounds for cVAF (two-tailed CIs for positive samples and one-tailed upper bound for negative samples). This is derived from the asymptotic distribution of this statistic $\chi^2$. We note that the CI does not overlap 0 in a sample that is called positive (VAF > 0).

## Simulation of detection rate for different error rates

We simulated theoretical performance given different error assumptions, depths, and tumor fractions. For a given number of 2K, 5K, 10K, 20K, and 40K variants and an expected depth in sequencing (10–100), we calculated the chance of seeing a molecule. Each molecule has a probability of being observed as a variant following two binomial distributions, one using the allelic fraction as probability and one using the error rate as probability. We performed 100,000 simulations for each combination of allelic fraction, depth, error rate, and evaluated the outcome using the MRD algorithm described above. Sensitivity was calculated as the number of positive predicted over the total number of simulations for each combination. Specificity was measured at ~99% using an allelic fraction of 0 as input, as expected from the MRD model parameter settings.

## Analytical sensitivity and specificity calculation using contrived samples

For the healthy titration experiment, "test" cfDNA samples from three healthy individuals were independently spiked into one "background" cfDNA sample, to create three sets of serial dilution samples. Single nucleotide polymorphisms (SNPs) for this titration study were selected among autosomal and X-linked SNPs using the following criteria: (1) Sites had to be heterozygous in the "test" individual and absent in the "background" individual; (2) Sequencing depth was required to be between 20× and 100× at all variable sites for all individuals; (3) The observed VAF had to be between 0.4 and 0.6 in the undiluted healthy cfDNA samples for the selected sites; (4) SNPs from repeat regions, as well as C->T variants at CpG sites were excluded; (5) SNPs present in multiple plasma samples were excluded.

This resulted in between 124K and 165K SNVs per sample. These SNVs were used as markers for the presence of the test sample in the dilution. Depth of sequence and presence of the donor allele was assessed at each of the target sites. For each plasma titration sample, we randomly subsampled 2K, 5K, 10K, or 20K variants constrained to having the same distribution of mutation types as in a typical CRC tumor. The variant profiles representative of CRC tumors were obtained by first collecting variant counts for each variant type from tissue WGS data of each CRC patient, and then calculating the fraction of each variant type among the total somatic SNVs. The fraction of each variant type for each patient was also calculated and the distributions of frequencies were used to obtain the standard deviations. The CRC mutation profile was shown in Appendix Fig. S2. 1000 rounds of subsampling were obtained at each variant level for every plasma titration sample, and the MRD algorithm was run on each of them. Sensitivity was measured as the rate at which subsamples were called as positive. For the specificity analysis, we generated random mutations at locations adjacent to those of the germline variants used in the sensitivity analysis, as long as they did not result in CpG sites. We randomly subsampled 2K, 5K, 10K, or 20K variants, also constrained the subsamples to have the same distribution of mutation types as in a typical CRC tumor. The MRD algorithm was run on each of the 1000 subsamples generated for each plasma

titration. Specificity was measured as the fraction of subsamples that resulted in a negative MRD call.

For the cancer titration experiment, we diluted a plasma sample from a melanoma cancer patient into a serum sample from a healthy individual to different concentration levels ranging from $1 \times 10^{-3}$ to $2 \times 10^{-6}$. We then sequenced the contrived plasma samples and ran the MRD analysis using the variant list from the melanoma tumor sample.

## AccuScan clinical specificity

To assess the performance of the assay, we evaluated the specificity of our assay in patient plasmas using tumor variants found in mismatched patient tumors. A total of ~1.3 M tumor (somatic) variants were collected from 57 patients, including 32 CRC, 17 ESCC and 8 melanoma patients. A summary of all patients and samples used in different data analysis were listed in Dataset EV1. We imposed all conditions required in tumor-variant-selection. For each plasma sample, we first excluded variants that were observed in the tissue or WBC BAMs of the corresponding patient. In each case, we sampled the number of variants depending on the average depth in the plasma sample, so that the expected sum of depth was equivalent to sampling 2K, 5K, 10K, or 20K variants in a sample with an average depth of 60. The number of variants sampled ($v_i$) for a given plasma sample with depth $dp_i$ and a target variant count of $n$ is:

$$v_i = \frac{60}{dp_i} \cdot n, \text{ where } n = \{2K, 5K, 10K, 20K\}$$

For example, to simulate a clinical sample with equivalent variant count of $n = 2K$ in a plasma sample that has an average depth of 30, the number of sampled variants out of 1.3 M possible somatic variants is 60/30*2000 = 4000.

For each plasma and equivalent number of variants, we generated 2000 random subsampled variant lists and ran the MRD test with each list using a nominal specificity of 99%. For each plasma sample, we measure the specificity as 1 minus the fraction of subsamples with a falsely positive outcome for the 2000 simulated MRD tests analyzed with mismatched tumor variant lists:

$$\text{Specificity}_i = 1 - \frac{FP_i}{2000}$$

## Statistical analysis

Data visualization was done by R (4.2.2, The R Foundation for Statistical Computing, Vienna, Austria, https://www.r-project.org/), statistical analysis was done using Python (3.9, Python Software Foundation, https://www.python.org/). The two-sided test $P < 0.05$ shows statistical significance.

## For more information

https://www.youtube.com/watch?v=XOcmGYtxND4; https://www.youtube.com/watch?v=eemsYPFChAg; https://www.youtube.com/watch?v=5MF2En3bCTY&t=513s.

## The paper explained

### Problem

Timely and accurate measurement of molecular residual disease (MRD) is critical for recurrence risk assessment and treatment decisions. Whole genome sequencing (WGS) for circulating tumor DNA (ctDNA) detection is an emerging field, enabled by the recent drop in sequencing cost. While WGS has shown feasibility for MRD detection, its wide use has been hampered by low sensitivity and specificity due to the lack of an effective error correction mechanism at the whole genome scale.

### Results

In this study, we introduce AccuScan, a highly efficient cell-free DNA (cfDNA) WGS technology that significantly reduces the error rate by orders of magnitude. AccuScan demonstrates limit of detection at the parts per million (PPM) range in analytical studies and achieves high accuracy in predicting relapse in post-surgery patients with colorectal or esophageal cancers. In melanoma patients undergoing immunotherapy, AccuScan-measured ctDNA dynamics shows strong correlations with clinical outcomes. When combined with tumor sequencing, AccuScan offers a white blood cell-free workflow that greatly simplifies the process of tumor-informed molecular residual disease (MRD) detection.

### Impact

AccuScan is a novel cfDNA WGS technology featuring genome-wide error suppression at single-read level. It enables a white blood cell-free approach for identifying tumor-specific variants and detecting tumor-informed MRD. With its ultralow limit of detection, minimal sample requirement, rapid turnaround time and comprehensive molecular information, AccuScan holds great potential for a wide range of research and clinical applications.

## Data availability

The raw sequencing data generated in this work is not publicly available due to national legislation and patient privacy. The variant list produced in this study are available at Zenodo upon request to respect patient confidentiality and ensure that data usage complies with patient consent. (URL: https://zenodo.org/records/11188534). All other data supporting the study's findings are provided in the article and its supplementary information files. The computer code is available at Github (URL: https://github.com/AccuraGen/AccuScan-publication).

The source data of this paper are collected in the following database record: biostudies:S-SCDT-10_1038-S44321-024-00115-0.

## Peer review information

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

## Acknowledgements

This study was supported by the Shanghai Natural Science Foundation Project (21ZR1458200); Key talent introduction project of Tongji Hospital (2021); Clinical Research Incubation Program of Tongji Hospital [ITJ(ZD)2104]; and grants from the NIH Yale SPORE in Skin Cancer (NCI P50CA121974). Hongyan Wang and George Yeung led the AccuScan process.

## Author contributions

**Xinxing Li**: Resources; Data curation; Formal analysis; Investigation; Writing—original draft; Writing—review and editing. **Tao Liu**: Resources; Data curation; Formal analysis; Investigation; Writing—original draft; Writing—review and editing. **Antonella Bacchiocchi**: Resources; Data curation; Formal analysis; Investigation; Writing—original draft; Writing—review and editing. **Mengxing Li**: Resources; Data curation; Formal analysis; Investigation; Writing—original draft; Writing—review and editing. **Wen Cheng**: Resources; Investigation; Writing—review and editing. **Tobias Wittkop**: Software; Validation; Methodology; Writing—original draft; Writing—review and editing. **Fernando L Mendez**: Software; Validation; Methodology; Writing—original draft; Writing—review and editing. **Yingyu Wang**: Formal analysis; Validation; Writing—original draft; Writing—review and editing. **Paul Tang**: Resources; Writing—original draft; Writing—review and editing. **Qianqian Yao**: Data curation; Formal analysis; Visualization; Writing—original draft; Writing—review and editing. **Marcus W Bosenberg**: Resources; Funding acquisition; Investigation; Writing—review and editing. **Mario Sznol**: Data curation; Funding acquisition; Investigation; Writing—review and editing. **Qin Yan**: Resources; Data curation; Funding acquisition; Investigation; Writing—review and editing. **Malek Faham**: Conceptualization; Methodology; Writing—review and editing. **Li Weng**: Conceptualization; Resources; Validation; Methodology; Writing—original draft; Project administration; Writing—review and editing. **Ruth Halaban**: Resources; Data curation; Supervision; Funding acquisition; Project administration; Writing—review and editing. **Hai Jin**: Resources; Data curation; Supervision; Writing—original draft; Project administration; Writing—review and editing. **Zhiqian Hu**: Resources; Data curation; Supervision; Funding acquisition; Writing—original draft; Project administration; Writing—review and editing.

Source data underlying figure panels in this paper may have individual authorship assigned. Where available, figure panel/source data authorship is listed in the following database record: biostudies:S-SCDT-10_1038-S44321-024-00115-0.

## Disclosure and competing interests statement

QQY is an employee of Shanghai YunSheng Medical Laboratory Co., Ltd. LW, PT, FM, YYW, TW, and MF are employees or consultant of AccuraGen Inc. QY received research funding and speaker fee from AstraZeneca and is a Scientific Advisory Board member of AccuraGen, Inc. MWB received research funding from AstraZeneca. MS disclosed the following: stock options: Actym, Adaptive Biotechnologies, Amphivena, Asher, Evolveimmune, Intensity, Nextcure,

Normunity, Oncohost, Thetis; Stock: Johnson and Johnson, Glaxo-Smith Kline; consulting fees: Adagene, Adaptimmune, Agenus, Alkermes, Alligator, Anaptys, Asher, Astra Zeneca, Biond, Biontech, Boston Pharmaceuticals, Bristol-Myers, Dragonfly, Evaxion, Evolveimmune, Genentech-Roche, Gilead, Glaxo Smith Kline, Ichnos, Idera, Immunocore, Incyte, Innate pharma, Iovance, iTEOS, Jazz Pharmaceuticals, Kadmon-Sanofi, Kanaph, Merck, Molecular Partners, Nextcure, Nimbus, Normunity, Numab, Ocellaris-Lilly, Oncohost, Ontario Institute for Cancer Research, Partner Therapeutics, Pfizer, Pierre-Fabre, PIO Therapeutics, Pliant, Regeneron, Rootpath, Rubius, Sapience, Simcha, Stcube, Sumitomo, Targovax, Teva, Turnstone, Verastem, Xilio, all these companies are unrelated to this study. Other authors have no conflict of interest to declare.

# Expanded View Figures

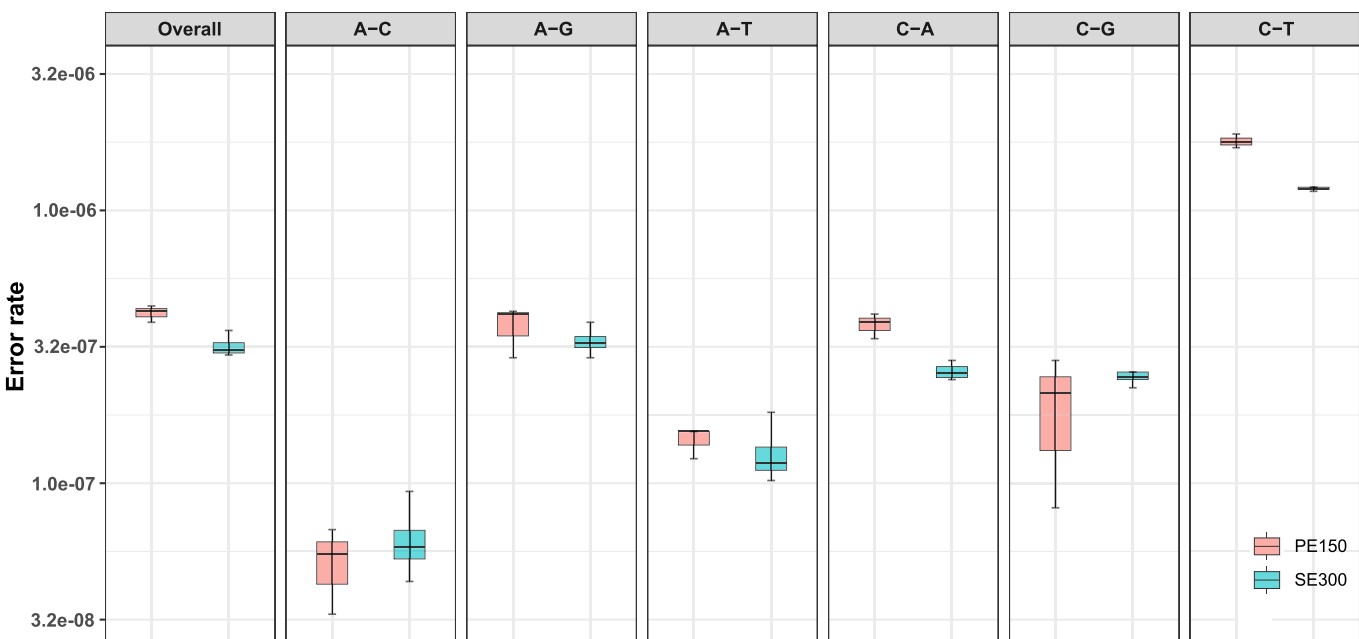

**Figure EV1. AccuScan error rate.**

Overall error rate and error rate of each variant type from AccuScan whole genome sequencing data on healthy human cell-free DNA samples ($n = 3$) sequenced by pair end 150 (PE150) or single end 300 (SE300) using the 300 cycle sequencing reagents. The line in the middle of the boxplot represents the median value; the box borders reflect the interquartile range (IQR, 25th to 75th percentiles); and the whiskers indicate 1.5 times IQR, with the lower line representing 25th percentile $- 1.5$*IQR and the upper line representing 75th percentile $+ 1.5$*IQR.

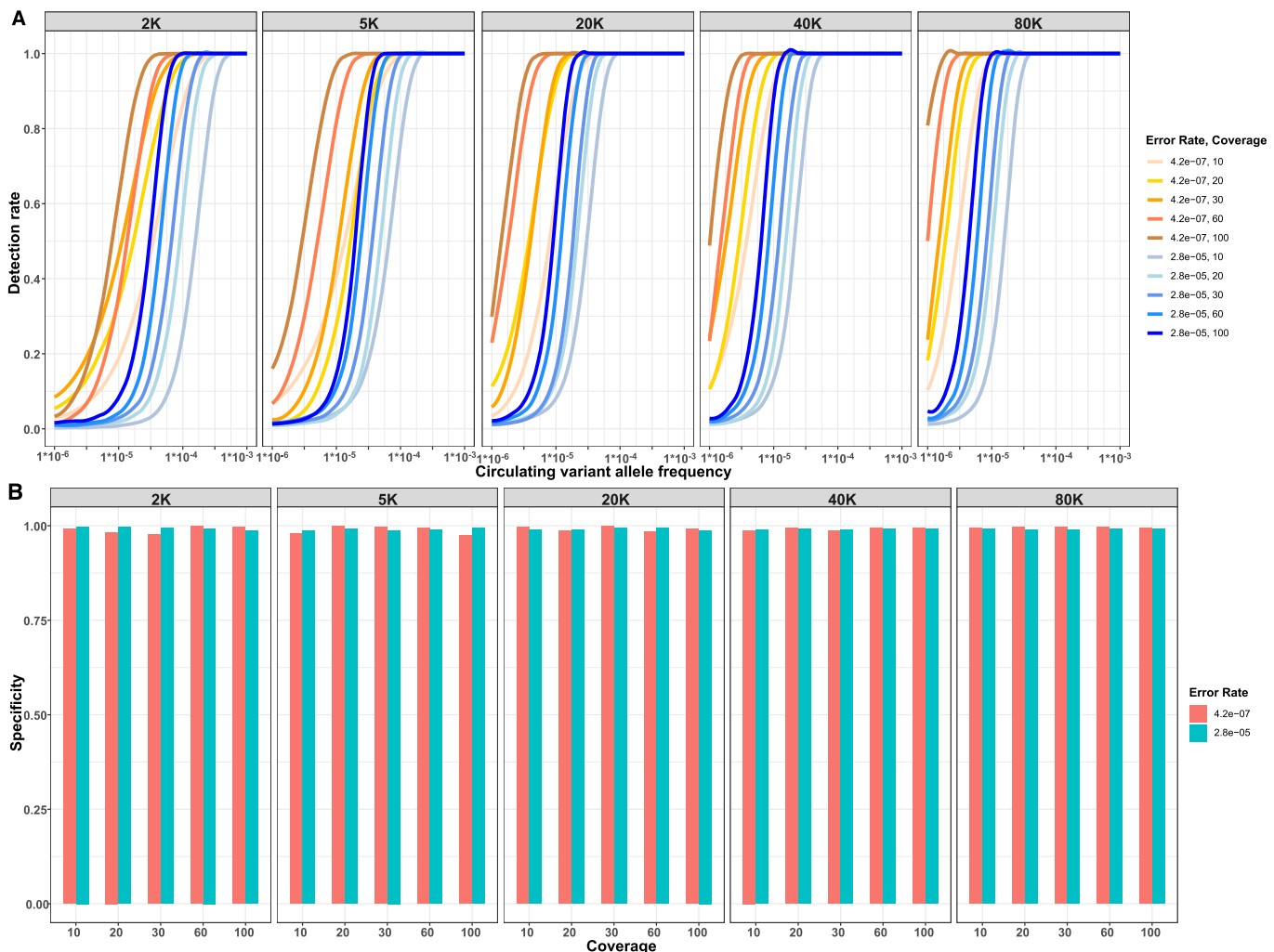

**Figure EV2. Analytical sensitivity and specificity of AccuScan.**

(A) Simulation using 2K, 5K, 20K, 40K, 80K markers and two different error rates to predict the theoretical detection rate under different sequencing coverages as a function of circulating variant allele frequency (cVAF). The $4.2 \times 10^{-7}$ error rate showed higher sensitivity than the $2.8 \times 10^{-5}$ error rate under the same sequencing depth. Detection rate is calculated as the fraction of tests that are called molecular residual disease (MRD) positive with the nominal specificity set at 99%. (B) Simulation using 2K, 5K, 20K, 40K, 80K markers and two different error rates to predict the theoretical specificity with the nominal specificity setting at 99%. Specificity is calculated as the fraction of tests that are called MRD negative when cVAF is zero.

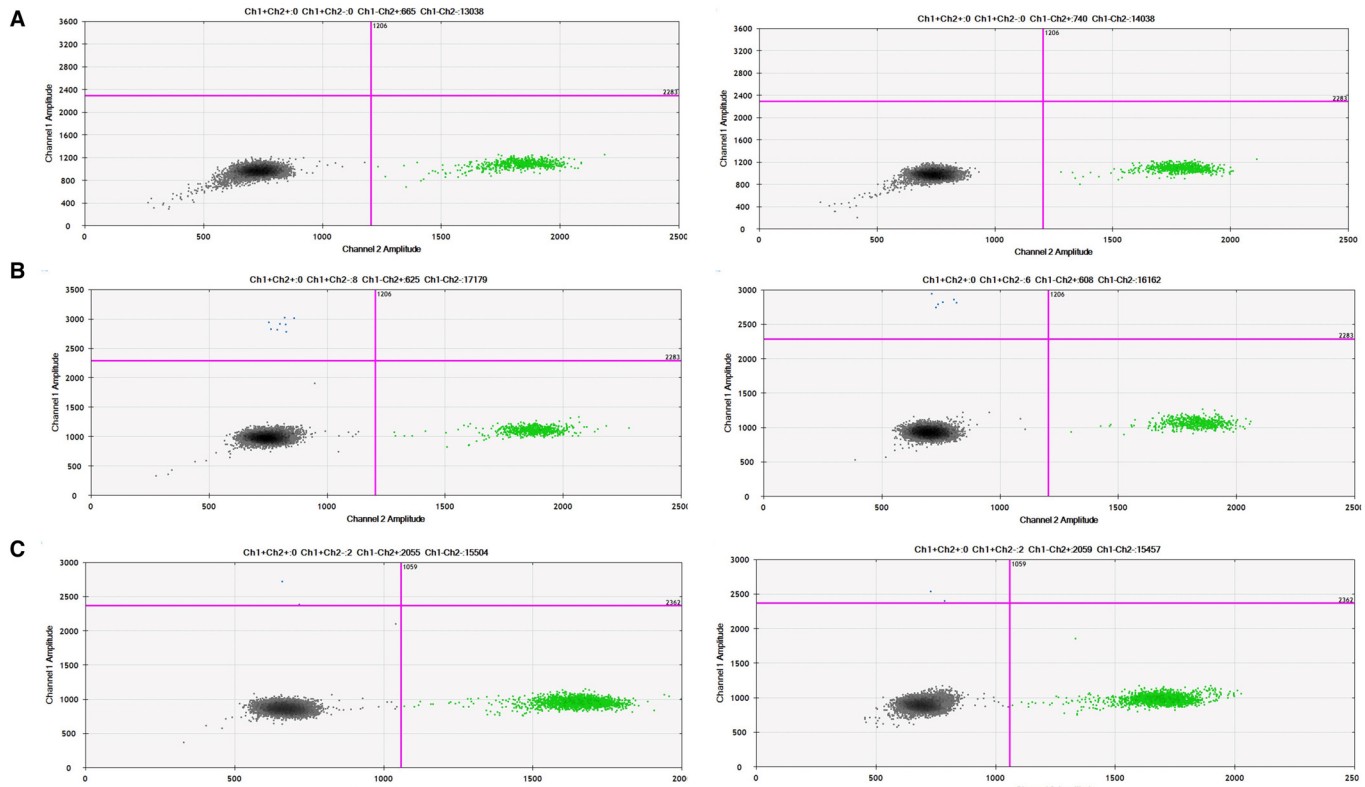

**Figure EV3. Droplet digital PCR of the melanoma cancer cell-free DNA sample.**

(A) Droplet digital PCR (ddPCR) analysis of a healthy plasma sample. (B) ddPCR analysis of the original melanoma cancer cell-free DNA (cfDNA) sample using the BRAF V600E assay. (C) ddPCR of the diluted melanoma cancer cfDNA sample in the healthy plasma background at an expected circulating variant allele frequency of 0.1%. Each shows the results of two independent replicates of ddPCR tests.

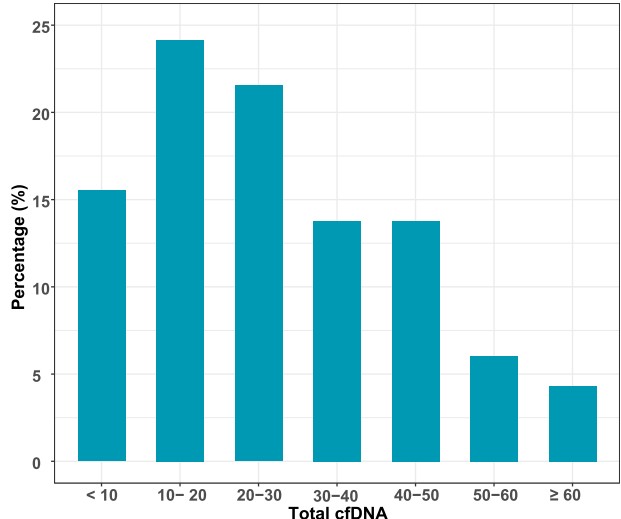

**Figure EV4.  Histogram of total cell-free DNA yield from plasma samples used in this study.**

Cell-free DNA (cfDNA) was extracted from 1 to 4 mL of plasma.

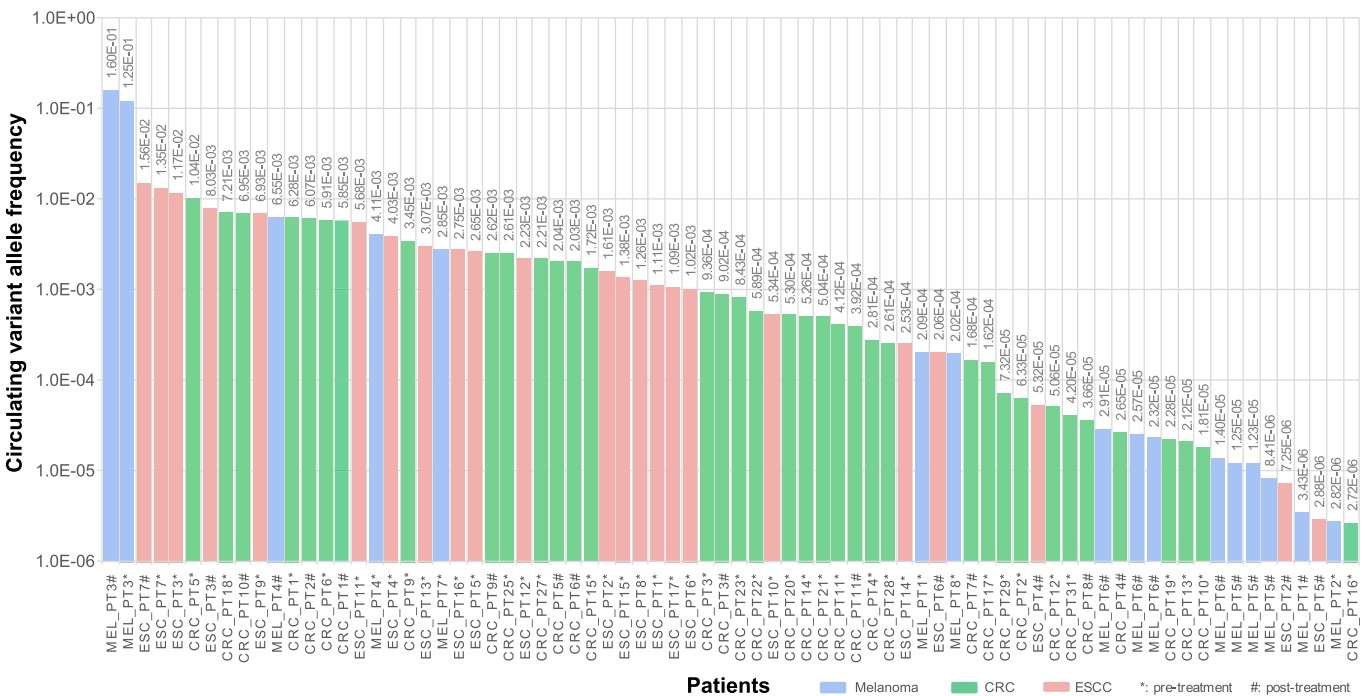

**Figure EV5. Circulating variant allele frequency of all circulating tumor DNA positive plasma samples.**

Plot included pre-treatment and post-treatment samples. CRC colorectal cancer, ESCC esophageal squamous cell carcinoma.

