## [Peer Review File · EMBO Molecular Medicine]

Ultra-sensitive molecular residual disease detection through whole genome sequencing with single-read error correction

Xinxing Li, Tao Liu, Antonella Bacchiocchi, Mengxing Li, Wen Cheng, Tobias Wittkop, Fernando Mendez, Yingyu Wang, Paul Tang, Qianqian Yao, Marcus Bosenberg, Mario Sznol, Qin Yan, Malek Faham, Li Weng, Ruth Halaban, Hai Jin, and Zhiqian Hu

Corresponding author(s): Li Weng (lweng@accuragen.com), Ruth Halaban (ruth.halaban@yale.edu), Hai Jin (jinhai@smmu.edu.cn), Zhiqian Hu (2105203@tongji.edu.cn)

Review Timeline:

Submission Date:	22nd Jan 24
Editorial Decision:	23rd Feb 24
Revision Received:	23rd May 24
Editorial Decision:	28th Jun 24
Revision Received:	6th Jul 24
Accepted:	16th Jul 24

Editor: Lise Roth

Transaction Report:

23rd Feb 2024

Dear Dr. Halaban,

Thank you for the submission of your manuscript to EMBO Molecular Medicine. We have now received feedback from the three reviewers who agreed to evaluate your manuscript. As you will see from the reports below, the referees acknowledge the interest of the study and are overall supporting publication of your work pending appropriate revisions.

Addressing the reviewers' concerns in full will be necessary for further considering the manuscript in our journal, and acceptance of the manuscript will entail a second round of review. EMBO Molecular Medicine encourages a single round of revision only and therefore, acceptance or rejection of the manuscript will depend on the completeness of your responses included in the next, final version of the manuscript. For this reason, and to save you from any frustrations in the end, I would strongly advise against returning an incomplete revision.

We are expecting your revised manuscript within three months, if you anticipate any delay, please contact us.

We require:

4) A .docx formatted letter INCLUDING the reviewers' reports and your detailed point-by-point responses to their comments. As part of the EMBO Press transparent editorial process, the point-by-point response is part of the Review Process File (RPF), which will be published alongside your paper.

5) A complete author checklist, which you can download from our author guidelines (<https://www.embopress.org/page/journal/17574684/authorguide#submissionofrevisions>). Please insert information in the checklist that is also reflected in the manuscript. The completed author checklist will also be part of the RPF.

6) Please note that all corresponding authors are required to supply an ORCID ID for their name upon submission of a revised manuscript.

7) It is mandatory to include a 'Data Availability' section after the Materials and Methods. Before submitting your revision, primary datasets produced in this study need to be deposited in an appropriate public database, and the accession numbers and database listed under 'Data Availability'. Please remember to provide a reviewer password if the datasets are not yet public (see <https://www.embopress.org/page/journal/17574684/authorguide#dataavailability>).

8) For data quantification: please specify the name of the statistical test used to generate error bars and P values, the number (n) of independent experiments (specify technical or biological replicates) underlying each data point and the test used to calculate p-values in each figure legend. The figure legends should contain a basic description of n, P and the test applied. Graphs must include a description of the bars and the error bars (s.d., s.e.m.). Please provide exact p values.

9) Our journal encourages inclusion of *data citations in the reference list* to directly cite datasets that were re-used and obtained from public databases. Data citations in the article text are distinct from normal bibliographical citations and should directly link to the database records from which the data can be accessed. In the main text, data citations are formatted as follows: "Data ref: Smith et al, 2001" or "Data ref: NCBI Sequence Read Archive PRJNA342805, 2017". In the Reference list,

data citations must be labeled with "[DATASET]". A data reference must provide the database name, accession number/identifiers and a resolvable link to the landing page from which the data can be accessed at the end of the reference. Further instructions are available at .

13) Author contributions: CRediT has replaced the traditional author contributions section because it offers a systematic machine readable author contributions format that allows for more effective research assessment. Please remove the Authors Contributions from the manuscript and use the free text boxes beneath each contributing author's name in our system to add specific details on the author's contribution. More information is available in our guide to authors.

16) As part of the EMBO Publications transparent editorial process initiative (see our Editorial at <http://embomolmed.embopress.org/content/2/9/329>), EMBO Molecular Medicine will publish online a Review Process File (RPF) to accompany accepted manuscripts.

In the event of acceptance, this file will be published in conjunction with your paper and will include the anonymous referee reports, your point-by-point response and all pertinent correspondence relating to the manuscript. Let us know whether you agree with the publication of the RPF and as here, if you want to remove or not any figures from it prior to publication. Please note that the Authors checklist will be published at the end of the RPF.

I look forward to receiving your revised manuscript.

Yours sincerely,

Lise Roth

**** Reviewer's comments ****

Referee #1 (Remarks for Author):

This manuscript by Li et al presents a new approach for detection of cfDNA called AccuScan. AccuScan takes denatured cfDNA and circularizes it, followed by whole-genome amplification to generate concatemer molecules containing tandem copies of the original template thus allowing for the same original cfDNA molecule to be covered by multiple sequencing reads. This allows for sequencing errors to be filtered. Authors report that AccuScan is an improvement on existing approaches for detection of cfDNA in molecular residual disease because it is more sensitive and specific, especially in cases of low input cfDNA. They demonstrated this in cfDNA samples for normal healthy donors, a melanoma patient with BRAF V600E mutation, CRC, ESCC and melanoma patients receiving immunotherapy.

Major comments:

1. This manuscript is not written in a way that is easy to understand. Extra effort should be made to improve readability.
 - a. For example, page 8: "To further confirm the sample level specificity of AccuScan, we collected more than 1.3M tumor-specific variants from 57 different cancer patients...randomly sampled a subset of 5K, 10K or 20K equivalent variants for testing the MRD call in mismatched patient plasma samples". What is meant by sample level specificity? And what is meant by collecting tumor-specific variants? What are "equivalent variants"? How can variants be equivalent or not equivalent?
 - b. Page 9: "...germline and CHIP mutations can be found at {greater than or equal to}2 molecules level, while tumor-specific variants will generally be found at single molecule level. Hence, we may remove variants found with 2 or more molecules in the post-treatment plasms from the tumor tissue sequencing result to obtain the list of tumor-specific mutations." What is meant by {greater than or equal to}2 molecules and single molecule?
 - c. Page 14: LOD95 not defined
 - d. Page 15: ACT not defined
 - e. Page 16: Authors quote a 45-56% landmark sensitivity for CRC with current commercial offerings. Where did they get this number? Reference?
2. Authors use cfDNA samples from healthy donors to estimate the average error rate. This is likely to be artificially lower than real world application of AccuScan for its intended use in detecting MRD from tumour cfDNA as there would be less noise and cfDNA fragmentation in healthy compared to cancer patients. This caveat should be included when discussing these results to caution the reader in the interpretation of these results.
3. It is certainly interesting to see AccuScan applied to real world patient samples in many different contexts: CRC, ESCC, melanoma and during immunotherapy treatment. While the results appear promising for detection of MRD, and authors even posit this approach may have a role in clinical decision making on whether to escalate or de-escalate adjuvant therapies, it is not clear how the data and analysis on clinical samples provided allows us to make a comparison between the AccuScan approach and other existing methodologies for MRD detection. The way it is reported here the reader cannot be led to the conclusion this approach offers any improvement over existing methods. Without making any comparisons, it does appear to be a sensitive and specific method and merits further investigation for potential clinical application.
4. What is missing is information on how the 58 subjects for inclusion in this study were selected. Was this a retrospective or prospective study? How many patients were consented, how many were excluded and were there criteria determining why these 58 patients were selected? This could have a profound impact on the results presented and skew the results to appear more sensitive for MRD detection than it would be in an unfiltered patient population.
5. How was intra-molecular ligation performed? Details on this are missing in the methods. Does this ligation process circularize all cfDNA molecules, or does this process alter the representation of cfDNA molecules compared to methods that do not require this step?

Minor comments:

1. Numerous grammatical errors, difficult to point out to the authors given no page/line numbers indicated:
 - a. Page 22, first and second sentences of section on "Analytical Sensitivity and Specificity calculation using contrived samples"
 - b. Subheadings inconsistently capitalized as sentence or initial case.

c. Page 12: "Patients 2"

2. Inconsistent numerical annotations:

- a. 20X and 100X some places, and 20x and 100x other places
- b. 5k, 10k, 20k some places, and 5000, 10000, 20000 other places

Referee #2 (Remarks for Author):

Li et al. report a WGS assay, AccuScan, for ctDNA-based detection of molecular residual disease (MRD). The assay utilizes rolling circle amplification (RCA) to control sequencing error rate, and couples it with a statistical framework for calling MRD status and estimating circulating tumor allele fraction (cTAF). The author report significantly reduced sequencing error rate from RCA compared to conventional WGS, and demonstrate high specificity and moderate to high sensitivity in MRD calls. The reported clinical applications are interesting and promising. The following methodology-related points should be clarified/addressed:

1. A key component of the assay is the use of RCA concatemers for error reduction. In Results, the authors state that a change from the reference that is consistent in all copies of concatemer is a presumed variant. In Methods, the authors define true variants as differences from the reference that were supported by at least two tandem copies within a read-pair. This inconsistency needs to be clarified.
2. For patients without matching white blood cell (WBC) samples, the authors tested and utilized a WBC-free workflow, which distinguishes tumor-specific variants from germline and CHIP mutations by defining the former as variants found at single molecule level and the latter as those found at {greater than or equal to}2 molecules levels. It is unclear how this can be achieved as there is no mention of tracking unique molecules in Methods. It is also unclear whether the cutoff is supported by any data.
3. In the results related to Figure 3A-D, the authors state that "With 40-60x sequencing of a low-tumor-burden (cTAF < 0.1%) plasma sample, germline and CHIP mutations can be found at {greater than or equal to} 2 molecules level, while tumor-specific variants will generally be found at single molecule level." By removing mutations with {greater than or equal to}2 molecules, they show that their methodology is comparable to a workflow that had additional sequencing of white blood cells. Thus, they claim that this methodology can thus be used in "place of WBC" sequencing. However, per their determination above regarding "low-tumor burden", it could be inferred that this may not be the case for patient with higher-tumor burden (for example, the BRAF-mt patient that had cTAF >1% in the preceding result sections). Is this true? If not, please clarify in the results section how their determination that their approach can be used in place of WBC is dependent or not-dependent on tumor burden.
4. The other key component of the assay is the statistical model for calling MRD status and estimating cTAF. The accuracy of cTAF estimation was assessed using a single BRAF mutation in a dilution experiment (Fig. 2E). Since a typical AccuScan assay involves tracking thousands of target mutations, the accuracy of cTAF estimation would be better assessed in such context, e.g. using the healthy sample mixtures described in the manuscript for assessing the performance of MRD calls, which involve 5,000 to 20,000 SNVs.
5. How many WGS SNVs do circulating tumor DNA and normal samples usually differ at? An estimation should be provided when presenting results from the selectivity analysis employed throughout the study to allow a proper interpretation of the significance of the analyses. Indeed, in a later section their sequencing "identified 5,820 tumor-specific variants per patient (2148-265800)". Because as few as ~2k variants can be identified in patients, the authors should provide measures of sensitivity/specificity to at least down to 2K variants throughout their different sensitivity analyses (instead of down to 5K variants as currently employed).
6. Regarding the sensitivity analysis (Fig 2C), the authors state that "SNVs were selected to have a variant-type profile similar to CRC tumors". Similarly, in the methods, they state they "also constrained the subsamples to having the same distribution of mutation types as in a typical CRC tumor". What explicit criteria was used to determine "variant-type profile"?

Referee #3 (Comments on Novelty/Model System for Author):

The methods section of the manuscript would benefit from more detail.

Referee #3 (Remarks for Author):

Review of the manuscript "Ultra-sensitive molecular residual disease detection through whole genome sequencing with single-read error correction" submitted by Xinxing Li et al for publication in EMBO Molecular Medicine

The authors present a novel approach for performing whole genome sequencing of plasma cell free DNA. The method includes several unique elements: 1) the input DNA is single stranded, 2) the DNA is circularized and amplified using rolling circle

amplification, which generates concatemers of the input DNA, 3) from the single concatemer'ed DNA WGS libraries are generated and sequenced using a standard library kit and sequencer. 4) due to the replicative nature of the DNA it is possible to do error correction at single read level, and the authors report that their approach achieves an error rate of 4.2×10^{-7} which is ~2 orders of magnitude lower than other approaches.

The authors use the new approach for MRD detection in the setting of colorectal and esophageal cancer. They furthermore use it for assessing response to immunotherapy in Melanoma patients.

While the approach is novel, promising and the results are interesting I do have some concerns. It is important that the authors acknowledge that it is just a proof-of-concept study and that a thorough large and biased study, where ctDNA analysis is performed blinded to the outcome of the patients are needed to appropriately assess the performance.

Major comments

Comment 1:

The authors should clarify whether the investigators were blinded to the outcome of the patients when doing the ctDNA and bioinformatic analyses?

Comment 2: On page 8 the Authors write

"To further confirm the sample level specificity of AccuScan, we collected more than 1.3M tumorspecific variants from 57 different cancer patients, including colorectal cancer (CRC), esophageal squamous cell carcinoma (ESCC) and melanoma, randomly sampled a subset of 5K, 10K or 20K equivalent variants for testing the MRD call in mismatched patient plasma samples. We did 2,000 random samplings of mismatched variants for each combination of variant count level and plasma sample. The average sample level specificity is computed as the fraction of MRD tests that are characterized with a negative MRD call. The observed values were similar to the nominal specificity, with 99.3%, 99.1%, and 98.9%, for 5K, 10K and 20K variant count levels, respectively. These results suggest that the AccuScan assay and analysis have the intended performance for patient plasmas using tumor variants (Data not shown)."

These results should be shown. The authors should also clarify whether they have matched tumor, WBC and plasma on these 57 patients. I assume they do the testing in 57 plasma samples, but this is not clear. Are the plasma samples preoperative or postoperative? What is the sensitivity in the paired analysis (tumor/plasma) of the 57 patients? I expect the fraction of positive MRD calls is higher when the mutations are matched to the plasma samples, than when they are mismatched. The authors should show this.

Comment 3: Section: "Identification of tumor-specific variants using a white blood cell free workflow"

Central to the proposed tumor-informed approach is the determination of the patient specific catalog of tumor-specific variants. To be able to filter out non-tumor specific variants a normal DNA control is needed. To avoid sequencing matched normal DNA the authors suggest to use data from a postoperative plasma sample as normal control, instead. They motivate this, by claiming the the cTAF in such a sample will be very low (<0.1%) and with 40-60x sequencing of the plasma DNA, then all variants (but those occurring just once) are likely to be germline variants, which should be filtered from the tumor variant catalog. The authors claim they also filter CHIP variants by this approach. However, this not completely true. They only filter CHIP variants if their plasma allele frequency is very high >3% (2/60). CHIP variants with plasma allele frequencies below this level will not be filtered. This paper <https://doi.org/10.1038/s41591-019-0652-7> indicates that the vast majority of CHIP variants (variants found in plasma and WBC, but not in the tumor) have plasma VAFs below 3% - most actually have plasma allele frequencies below 0.2%. The authors should revise the text to reflect this.

In the section the authors write "AccuScan MRD analysis of the plasma samples (n=48) from these 20 patients returned identical MRD calls under either workflow". The authors should explain which 20 patients and how they get to 48 plasma samples. Are the 48 samples all collected post-operatively or do the authors include preoperative plasma as well.

Comment 4: Tables S3-S5

The authors should provide sufficient clinical information to reproduce the results. This includes adding outcome event information to Tables S3, S4, S5

For tables S3 and S4 this includes

- Event status (no event, recurrence, death)
- Time to event (or censoring)
- Adjuvant chemotherapy status

For Tabel S5 the information should include the response evaluations and the progression free survival

Comment 5: Section MRD detection and prognostic value in surgical patients

The authors should specify the outcomes and results for the patients with an available landmark sample. It is confusing that they do not report specifically on the landmark.

They provide results for an analysis including all available postOP samples. This latter analysis may well be confounded/biased as it is clear from the swimmerplots in Figure 3 that the non-landmark samples were not systematically collected. For the CRC recurrence patients non-landmark samples are only available when the landmark was falsely negative or not available.

The authors should report the landmark results separately for a unbiased and clearer presentation.

Comment 6: Discussion section

The authors provide present data and results in the discussion section (Figure S5). This is inappropriate. The authors should

present the results of Fig S5 in the results section.

Comment 7: Discussion section

The reported results can only be seen as proof-of-principle. Nowhere in the manuscript is there a description of the process and criteria for including patients. The patients and samples seem to have been highly selected. For instance the CRC cohort, the recurrence cases are seemingly much younger than the non-recurrence patients (Table S3). Furthermore, from Figure 3, one can see the collection time-points of the landmark samples are also biased. For the recurrence patients the landmark samples were collected at between month 1 and 2, while it for the majority of the non-recurrence patients were collected within a week or two of surgery. This difference could indicate the samples were not collected as part of the same study and maybe not using the same procedure either. We cannot tell as the authors do not provide sufficient detail.

While this is suboptimal, I don't see it as a major problem for a proof of principle paper. Though, one should be careful not to compare the sensitivity and specificity obtained from such biased cohorts with studies reporting on consecutive non-selected cohorts where the samples were collected using the standardized procedures.

Therefore I strongly recommend the authors to reformulate or remove the following statement in the discussion "Considering an average sensitivity of 45%-56% of landmark sensitivity for CRC with current commercial offerings, AccuScan picked up as much as twice the number of patients who may benefit from adjuvant therapy without increasing over treatment."

For the comparison to make sense, the AccuScan performances need to be confirmed in independent, large, and consecutively collected patient cohorts. Importantly, the AccuScan investigators should perform these analyses blinded to the patient outcome.

Comment 8: Experimental Design

The authors should explain where and when the patients were recruited. They should also explain and justify the apparently biased variation in the landmark sampling time-points of the CRC cohort.

Comment 9: Section Data and materials availability.

The authors claim that the cfDNA library preparation and variant calling using concatemers have been previously reported in detail (Refs 38 and 45). However, these references are insufficient. In ref38 - the mat and met sections refers back to another paper for description of how CLamp-seq is performed. In ref 45, the methods used are only described in the supplementary materials and only very briefly.

I highly recommend the authors to provide sufficiently detailed method information in the present manuscript, to allow the readers to reproduce the results.

For example in the section "AccuScan error correction algorithm and error rate measurement" the authors write "Variant calls using concatemer has been previously described in (45). Summarizing, reads were aligned to the reference genome (hg38) and high read quality single nucleotide substitutions different to the reference are further investigated for repeat confirmation." This is insufficient, they need to specify the definition of "high read quality single nucleotide substitutions"

Comment 10: Section Data and materials availability.

Related to the previous comment. The authors should make available the code used in the paper - this includes the error rate assessment and MRD algorithm codes. This is central for the readers and reviewers to be able to assess the appropriateness and quality of the used procedures. This is also central for others to be able to confirm and hopefully independently reproduce the results.

I understand that the raw sequencing data cannot be shared, however the variant lists should not be sensitive and should be made available for all patients/samples (before and after filtering).

The code should make it possible for readers of the paper to reproduce the results from the variant call files.

A table should be provided with all patients and samples, including an indication of what samples were included in which analyses (preoperative, landmark, longitudinal etc).

Comment 11: Section AccuScan error correction algorithm and error rate

How dependent are the estimated error rates of the 200Mb selected positions?

The authors should repeat the selection of the 200 Mb positions and report the error rates with variance estimates.

The authors should motivate why the C to T variants at CpG sites were excluded.

The authors write "Within a given plasma sample, we removed variant positions where two or more variant molecules"

However, re-occurring errors are still errors and should be included when reporting the error estimates of the AccuScan seq approach.

If the authors in practice always filter out variants with 2 or more cfDNA molecules detected in plasma, then they should motivate this, and explain why re-occurring errors should be removed. Then it would also be justified to also show the error rates

based on this procedure.

Comment 12: Section AccuScan MRD algorithm

The text reads "The number of tumor base calls was modelled as a random variable that depends on the sequencing depth, error rates, and tumor allele frequency"

With the reference to tumor allele frequency do the authors refer to the allele frequency observed in the tumor or to the circulating tumor allele frequency (cTAF)

This should be explained.

The authors mention that the cTAF estimated by maximizing the likelihood of the observations with respect to the variant allele frequency and the error rate. If they refer to the variant allele frequency in circulation it would be appropriate to label it cVAF.

Comment 13: Section AccuScan MRD algorithm

In the error rate section, the authors explained that they filter out all variants for which 2 or more reads are available when estimating the cfDNA error rate.

However, it is unclear to me if this is also done, when counting how many cfDNA fragments (observations) of a given variant type are in patient plasma samples (the Cv's).

This should be specified.

Comment 14: procedure for calling plasma samples ctDNA positive/negative

The authors should motivate the cut-off they have defined for calling a sample ctDNA positive or not. They write that they reject the null hypothesis when the cumulative distribution of a χ^2 distribution with 1 degree of freedom evaluated at the difference in log-likelihoods is greater 0.98"

What is the 0.98?

Moreover, the authors should explain when this cut-off was defined. Was it done before the study was initiated and blinded to patient outcome. Was it changed during the study to get the optimal separation between recurrence and non-recurrence patients?

I will recommend the authors to acknowledge in the manuscript, that to get a confident assessment of AccuScan performance at the landmark time point for colorectal and esophageal cancer, then the approach should be locked (the lab approach, the presented statistical procedure and the threshold for calling the samples) and applied to an independent unbiased patient cohort (or cohorts) with at least 3- years of follow-up.

Comment 15: cTAF 95% confidence bounds

The authors write "We obtained 95% confidence bounds for our cTAF estimates: two-tailed confidence intervals for positive samples and one-tailed upper bound for negative samples.

The approach used for this should be explained in detail.

Comment 16: the confidence in the ctDNA calls and cTAF estimates

The confidence in the ctDNA calls and cTAF estimates made by the AccuScan approach will be much improved if the authors run an well-established ctDNA detection strategy (e.g. digital PCR) on a subset of patients/samples (paired sample aliquot) and show the agreement between the two approaches in ctDNA calls and estimated cTAFs.

Minor comments:

1) In the abstract the authors should specify

a) the definition of the "landmark",

b) the performance of the approach for detection MRD in Esophageal cancer

c) what is meant by predictive value for immunotherapy monitoring

2) Fig S2: There are two panels in A), B) and C). It is not explained what they reflect.

3) The citation provided for Ref 38 is incomplete

4) Section "AccuScan MRD algorithm". The authors write "We estimate the cTAF by maximizing the likelihood of the observations with respect to the variant allele frequency (VAF , $VAF \geq 0$) and the error rate". What is meant by (VAF , $VAF \geq 0$) ?

Manuscript title: Ultra-sensitive molecular residual disease detection through whole genome sequencing with single-read error correction

Manuscript ID: EMM-2024-19339

Dear editor and reviewers,

We thank all the reviewers for their constructive comments, which are valuable for improving the quality of our work. We have carefully reviewed each comment and incorporated the suggestions into the revised manuscript. The responses to the reviewer's comments and key revisions are outlined as follows. The reviewers' comments are in black text, accompanied by our responses in blue text, the corresponding revisions highlighted in red for clarity, and the references for our responses are listed at the end.

(The number of lines and pages marked below in this letter indicate the corresponding position of the modification in the clean version of revised manuscript.)

Response to Reviewers' comments

Referee #1:

This manuscript by Li et al presents a new approach for detection of cfDNA called AccuScan. AccuScan takes denatured cfDNA and circularizes it, followed by whole-genome amplification to generate concatemer molecules containing tandem copies of the original template thus allowing for the same original cfDNA molecule to be covered by multiple sequencing reads. This allows for sequencing errors to be filtered. Authors report that AccuScan is an improvement on existing approaches for detection of cfDNA in molecular residual disease because it is more sensitive and specific, especially in cases of low input cfDNA. They demonstrated this in cfDNA samples for normal healthy donors, a melanoma patient with BRAF V600E mutation, CRC, ESCC and melanoma patients receiving immunotherapy.

We thank the reviewer for the summary.

Major comments:

1. This manuscript is not written in a way that is easy to understand. Extra effort should be made to improve readability.

We thank the reviewer for the careful review. We have revised the manuscript extensively to improve the readability. Please find the revisions below.

a. For example, page 8: "To further confirm the sample level specificity of AccuScan, we collected more than 1.3M tumor-specific variants from 57 different cancer patients...randomly sampled a subset of 5K, 10K or 20K equivalent variants for testing the MRD call in mismatched patient plasma samples". What is meant by sample level specificity? And what is meant by collecting tumor-specific variants? What are "equivalent variants"? How can variants be equivalent or not equivalent? Please explain "sample level specificity", "collecting tumor-specific variants", "equivalent variants" details in the supplementary material and methods; revise the results section to make it easy for understanding.

Thank you for raising the concern that the specificity analysis description should be clearer. We added more information in the manuscript.

"Sample-level specificity" refers to the ratio of negative MRD calls to all the MRD tests performed. This term was defined in contrast to specificity of individual variants. In this experiment, the plasma samples were assessed using mis-matched tumor variants, therefore all the MRD calls were expected to be negative, and any observed MRD positive calls were considered as false positives.

"Tumor-specific variants" in this context are somatic variants detected in the patient tumor-tissue samples in this manuscript. We changed the description to clarify the definition of the variants used in this analysis.

"Equivalent variants" should be "equivalent variant count". It defines the number of tumor variants used for MRD detection under the condition that the plasma sample has an average sequencing depth of 60x. The specificity analysis aimed to assess all plasma samples with the same number of total cfDNA fragments examined, which is the average depth times the number of variants. However, not all cfDNA samples

were sequenced to the exact same depth, we normalized the number of selected variants to a depth of 60× in our sampling. From the pool of ~1.3 million variants, we randomly sampled subsets equivalent to 2K, 5K, 10K, or 20K variants at a depth of 60x to test MRD calls in mismatched patient plasma samples. For example, if a plasma sample has an average depth of 30×, sampling 10K variants would be necessary to match an equivalent variant count of 5K at a depth of 60×. We added more information in the manuscript to describe this in detail.

We have revised the manuscript as following:

Revision in Result section:

To further corroborate the specificity of AccuScan, we utilized over 1.3 million tumor variants from tumor tissue samples and simulated datasets of variants to be tested in plasma samples from mismatched individuals. The MRD model and its nominal specificity of 99% was fixed before this test and did not rely on empirical data (Materials and methods). This analysis served the purpose of measuring the specificity of the MRD model in plasma samples from cancer patients. The specificity analysis aimed to assess all plasma samples using the same total number of cfDNA fragments, calculated as the average depth multiplied by the number of variants. Given that not all cfDNA samples were sequenced to the exact same depth, we normalized the number of selected variants to a depth of 60× in our sampling. From the pool of ~1.3 million variants, we randomly sampled subsets equivalent to 2K, 5K, 10K, or 20K variants at a depth of 60× to test MRD calls in mismatched patient plasma samples. For example, if a plasma sample has an average depth of 30×, sampling 10K variants would be necessary to match an equivalent variant count of 5K at a depth of 60×. We performed 2,000 random samplings of mismatched variants for each combination of equivalent variant count level and plasma sample. The specificity for each plasma sample at each equivalent variant count level is calculated as the number of negative MRD calls out of the total of 2,000 random samplings (Materials and methods). The observed average specificities are 99.6%, 99.3%, 99.1%, and 98.9%, for 2K, 5K, 10K and 20K equivalent variant count levels, respectively.

These results suggest that the AccuScan assay and analysis achieve the intended performance for patient plasma samples using tumor variants (Appendix Figure S5).

(Pages 9 - 10, lines 196 - 214)

Appendix Figure S5. The observed specificity shown as boxplots for each equivalent variant count number (2K, 5K, 10K, and 20K) for 117 mismatched plasma samples from 57 patients. The average specificity across all plasma samples is above the nominal specificity of 99% set in the model.

Revision in Materials and methods section:

AccuScan clinical specificity

To assess the performance of the assay, we evaluated the specificity of our assay in patient plasmas using tumor variants found in mis-matched patient tumors. Total ~1.3M tumor (somatic) variants were collected from 57 patients, including the 32 CRC, 17 ESCC and 8 melanoma patients. A summary of all patients and samples used in different data analysis were listed in Table EV9. We imposed all conditions required in tumor-variant-selection. For each plasma sample, we first excluded

variants that were observed in the tissue or WBC BAMs of the corresponding patient. In each case we sampled the number of variants depending on the average depth in the plasma sample, so that the expected sum of depth was equivalent to sampling 2K, 5K, 10K or 20K variants in a sample with an average depth of 60. The number of variants sampled (v_i) for a given plasma sample with depth dp_i and a target variant count of n is:

$$v_i = \frac{60}{dp_i} \cdot n, \text{ where } n = \{2K, 5K, 10K, 20K\}$$

For example, to simulate a clinical sample with equivalent variant count of $n = 2K$ in a plasma sample that has an average dp of 30, the number of sampled variants out of 1.3M possible somatic variants is $60/30 \cdot 2000 = 4000$.

For each plasma and equivalent number of variants, we generated 2000 random subsampled variant lists and ran the MRD test with each list using a nominal specificity of 99%. For each plasma sample, we measure the specificity as 1 minus the fraction of subsamples with a falsely positive outcome for the 2,000 simulated MRD test analyzed with mismatched tumor variant lists:

$$\text{Specificity}_i = 1 - \frac{FP_i}{2000} \text{ (Pages 29 - 30, lines 639 - 658)}$$

b. Page 9: "...germline and CHIP mutations can be found at {greater than or equal to}2 molecules level, while tumor-specific variants will generally be found at single molecule level. Hence, we may remove variants found with 2 or more molecules in the post-treatment plasms from the tumor tissue sequencing result to obtain the list of tumor-specific mutations." What is meant by {greater than or equal to}2 molecules and single molecule?

We thank the reviewer for pointing out the confusion. The number of molecules here refers to cfDNA molecules. cfDNA molecules in plasm have been shown to be

fragmented by endogenous DNases (Serpas et al, 2019; Snyder et al, 2016). Although the physiological fragmentation process is not entirely random, it is unlikely that two ctDNA molecules will have the exact same breakpoints when the sequencing depth is low. We therefore use the start and end positions of a cfDNA molecule as the unique identifier to count the number of unique molecules (Material and methods). We have revised the manuscript to clarify the method.

Revision in Results section:

We used the start and end positions of a cfDNA molecule as the unique identifier to count the number of unique molecules (Materials and methods) (Page 9, lines 189 - 190)

With 40-60× sequencing of cfDNA extracted from a low-tumor-burden (cVAF < 0.1%) plasma sample, germline SNVs (cVAF = 50% or 100%) are typically observed at ≥ 2 unique cfDNA molecule levels, while tumor-specific variants (cVAF < 0.1%) are generally found at < 2 unique cfDNA molecule levels (Appendix Figure S6). Therefore, we may remove variants found with 2 or more molecules in the post-treatment plasmas from the tumor tissue sequencing results to obtain the list of tumor-specific mutations. (Page 11, lines 231 - 236)

Appendix Figure S6. Distribution of probability and expected number of observed variants in Poisson model at different allele frequencies.

A Probability distribution of Poisson model using depth of 40 and circulating variant allele frequencies (cVAF) ranging from 0.1% to 50%. X-axis is the number of observed variant molecules and y-axis is the probability to see this many variant molecules for a given variant for different cVAFs.

B Similar to A but showing the expected number of observed variants under the assumption of evaluating 5K variants as tumor-specific markers.

Revision in Materials and methods section:

First, reads were aligned to the reference genome (hg38) using bwa mem (Li, 2013). Molecule boundaries are determined from neighboring alignments within the same read-pair and can be used for quantification. Molecule depth is calculated at each position and requires repeat confirmation for variant and wild-type calls. **(Page 24, lines 525 - 529)**

c. Page 14: LOD95 not defined

We thank the reviewer for pointing out this. We defined LOD95 as limit of detection with 95% probability of detecting the presence of ctDNA.

Revision in Discussion section

AccuScan reduces the SNV error rate to less than 1 in 2 million with PE150 sequencing, enabling LOD with 95% probability (LOD95) of detecting tumor-specific mutations at 50 PPM with 10× coverage using 10K SNV markers, and detection of tumor-specific mutations down to ≤ 10 PPM at 60× coverage, while maintaining a high sample-level specificity of 99%. **(Page 18, lines 388 - 392)**

d. Page 15: ACT not defined

We thank the reviewer for the comment. We have changed ACT to adjuvant chemotherapy in the manuscript as following:

Revision in Discussion section:

For example, National Comprehensive Cancer Network guideline recommends starting adjuvant chemotherapy for high-risk CRC patients no later than 6-8 weeks

after surgery (Benson et al, 2021). (Page 19, lines 418 - 420)

e. Page 16: Authors quote a 45-56% landmark sensitivity for CRC with current commercial offerings. Where did they get this number? Reference?

We thank the reviewer for pointing out the missing information. We have added three references to support these numbers.

Current commercial offerings showed an average sensitivity of 41%-56% of at landmark for CRC (Parikh et al, 2021; Reinert et al, 2019; Tarazona et al, 2019). (Page 19, lines 421 - 423)

Reference

(1) Reinert T, Henriksen TV, Christensen E, Sharma S, Salari R, Sethi H, Knudsen M, Nordentoft I, Wu HT, Tin AS et al (2019) Analysis of Plasma Cell-Free DNA by Ultradeep Sequencing in Patients With Stages I to III Colorectal Cancer. *JAMA Oncol* 5: 1124-1131

The 41.2% (7/17) sensitivity at landmark (day 30 after surgery) was calculated using data extracted from eFigure 6.

(2) Parikh AR, Van Seventer EE, Siravegna G, Hartwig AV, Jaimovich A, He Y, Kanter K, Fish MG, Fosbenner KD, Miao B et al (2021) Minimal Residual Disease Detection using a Plasma-only Circulating Tumor DNA Assay in Patients with Colorectal Cancer. *Clin Cancer Res* 27: 5586-5594

This article reported 55.6% (15/27) sensitivity at landmark (one month after surgery, Supplementary Figure S1A)

(3) Tarazona N, Gimeno-Valiente F, Gambardella V, Zuniga S, Rentero-Garrido P, Huerta M, Rosello S, Martinez-Ciarpaglini C, Carbonell-Asins JA, Carrasco F et al (2019) Targeted next-generation sequencing of circulating-tumor DNA for tracking minimal residual disease in localized colon cancer. *Ann Oncol* 30: 1804-1812

This study reported 47.1% (8/17) sensitivity at landmark (6-8 weeks after surgery), as described in the article: “In 17 patients relapsing during follow-up, 8 (47.1%) ctDNA had already been detected in the first plasma sample taken after surgery.”

2. Authors use cfDNA samples from healthy donors to estimate the average error rate. This is likely to be artificially lower than real world application of AccuScan for its intended use in detecting MRD from tumour cfDNA as there would be less noise and cfDNA fragmentation in healthy compared to cancer patients. This caveat should be included when discussing these results to caution the reader in the interpretation of these results.

We thank the reviewer for the insightful suggestion. We have added the content in the discussion section as you recommended.

Revision in Discussion section

It is worth noting that the error rates shown in Figure 1B were measured using plasma samples from three healthy individuals, aiming to compare regular WGS method with AccuScan. It is possible that the error rate observed in a cancer patient sample may be higher than those in the healthy plasma samples. Therefore, it is important to calculate the error rate for each cancer patient sample when applying AccuScan to MRD detection to accurately assess noise at the individual level. (Page 18, lines 399 - 404)

3. It is certainly interesting to see AccuScan applied to real world patient samples in many different contexts: CRC, ESCC, melanoma and during immunotherapy treatment. While the results appear promising for detection of MRD, and authors even posit this approach may have a role in clinical decision making on whether to escalate or de-escalate adjuvant therapies, it is not clear how the data and analysis on clinical samples provided allows us to make a comparison between the AccuScan approach and other existing methodologies for MRD detection. The way it is reported here the reader cannot be led to the conclusion this approach offers any improvement over existing methods. Without making any comparisons, it does appear to be a sensitive and specific method and merits further investigation for potential clinical application.

Thanks for your valuable comments. We agree with you that this is a proof-of-principle study. Data presented in this study is preliminary and is not sufficient to support clinical decisions based on the AccuScan ctDNA MRD calls. As a sensitive and specific MRD detection method, AccuScan, like other MRD tests, has

the potential to guide adjuvant therapy in CRC patients in the future. Large clinical trials and interventional studies are needed to validate the clinical utility of AccuScan. We have revised the manuscript based on the reviewer's comment.

Revision in Discussion Section

In this report, AccuScan demonstrated analytical sensitivity at PPM level with contrived samples and achieved a landmark sensitivity of 90% for predicting relapse after surgery with 100% specificity in retrospective CRC patient samples. Taken together, these results indicate that AccuScan test has the potential to achieve very high sensitivity and specificity for MRD detection. It is important to note that the data presented here are from proof-of-principle studies. Large, blinded clinical studies with consecutively collected patient cohorts and sufficient follow-up (i.e., ≥ 3 years) are needed to confirm and validate the clinical performance of AccuScan. (Pages 19-20, lines 428 - 435)

4. What is missing is information on how the 58 subjects for inclusion in this study were selected. Was this a retrospective or prospective study? How many patients were consented, how many were excluded and were there criteria determining why these 58 patients were selected? This could have a profound impact on the results presented and skew the results to appear more sensitive for MRD detection than it would be in an unfiltered patient population.

We thank the reviewer for the comment. This was a retrospective study. CRC and ESCC cancer patients met the following criteria were included for the MRD study: 1) Patient had surgery and primary tumor tissue samples were available; 2) Blood samples collected after surgery with ≥ 1 mL of plasma available; 3) Patient had follow-up till relapse or death or minimum of 18 months. Melanoma patients who met the following criteria were included for the immune therapy monitoring study: 1) Tumor tissue samples available; 2) Patient had at least two blood samples collected during the immunotherapy, with ≥ 2 mL of plasma available at each time point; 3) Patient had follow-up information till relapse or minimum of 12 months. We have

added the inclusion criteria (Table EV8) in the revised manuscript.

Revision in Materials and methods section

Participants were recruited in four hospitals. The CRC patients' samples were collected from Shanghai Changzheng Hospital (IRB: CZ2017-251) and Tongji Hospital (IRB: K-KYSB-2021-005) and from July 2017 to February 2023. The ESCC patients' samples were collected from Shanghai Changhai Hospital (IRB: CHEC2020-021) from June 2020 to December 2021. The melanoma patients' samples were collected from Yale School of Medicine (IRB: 0609001869) from March 2019 to February 2023. A total of 58 subjects including 32 CRC patients, 17 ESCC patients and 9 melanoma patients were included in this report. Inclusion and exclusion criteria were listed in Table EV8. (Page 21, lines 460 - 467)

5. How was intra-molecular ligation performed? Details on this are missing in the methods. Does this ligation process circularize all cfDNA molecules, or does this process alter the representation of cfDNA molecules compared to methods that do not require this step?

We thank the reviewer for the comment. We have added detailed information in the Material and methods section. The intramolecular ligation was performed by denaturing the cfDNA and directly circularizing the single stranded DNA molecules using a single strand DNA ligase, as described in the previous study (Wang et al, 2024; Wang et al, 2020; Xu et al, 2017). The intra-molecular ligation process can effectively circularize cfDNA molecules that are >15 bases in size. Unlike double strand library prep method which requires end repair before ligation, this method does not modify the natural ends of cfDNA and it captures short (<100bp) DNA molecules more efficiently. As a result, the cfDNA sequenced using this method showed a larger proportion of small fragments than standard double strand library preparation method, as shown in Figure 4A from a previous report (Wang et al, 2024). Figure 4A from Wang et al. illustrates the fragment size distribution of cfDNA samples sequenced with Circular Ligation Amplification and sequencing (CLAMP-seq) versus standard

double stranded sequencing. Frequency is the fraction of reads for each fragment size relative to all reads for each sample. Solid line: Fragment size distribution of a cfDNA sample from a subject with CRC prepared with single strand library (orange) and double strand library (blue). Dash line: Fragment size distribution of a cfDNA sample from a healthy subject sequenced with double strand library prep method (blue) and AccuScan single-stranded library prep method (orange). In both cases, the size of fragments sequenced through single strand DNA intra-molecular ligation shift towards left relative to the size observed through standard double strand DNA sequencing, suggesting that the single strand DNA intra-molecular ligation method captures smaller size fragments more efficiently than standard double strand DNA.

Figure 4A from Wang et al. 2024

Revision in Materials and methods Section

cfDNA library preparation was performed as described previously (Wang et al, 2024; Wang et al, 2020; Xu et al, 2017). Briefly, cfDNA molecules were heat denatured to become single-stranded and then circularized directly through intra-molecular ligation using a single-strand DNA (ssDNA) ligase, which is effective in circularize DNA molecules ≥ 15 bases in size (Xu et al, 2017). The circularized cfDNA molecules are then amplified by whole genome rolling circle amplification (RCA) using random hexamers as primers. The whole genome RCA products were sonicated into fragments with a peak at around 500 bp by ultrasonicator, followed by library construction using NEBNext® Ultra™ II Ligation Module (catalog no. E7595L, NEB) and NEBNext® Ultra™ II End Repair/dA-Tailing Module (catalog no. E7546L,

NEB). The cfDNA libraries were sequenced on DNBSEQ-T7 (MGI) with DNBSEQ-T7RS High-throughput Sequencing Set (FCL PE150). (Page 23, lines 501 - 511)

Minor comments:

Thank you for the thorough review and constructive suggestions. We have added the page and line numbers and revised the manuscript as you recommend.

1. Numerous grammatical errors, difficult to point out to the authors given no page/line numbers indicated:

a. Page 22, first and second sentences of section on "Analytical Sensitivity and Specificity calculation using contrived samples"

We have modified the first and second sentences of section on "Analytical Sensitivity and Specificity calculation using contrived samples" as following:

For the healthy titration experiment, “test” cfDNA samples from three healthy individuals were independently spiked into one “background” cfDNA sample, to create three sets of serial dilution samples. Single nucleotide polymorphisms (SNPs) for this titration study were selected among autosomal and X-linked SNPs using the following criteria: 1) Sites had to be heterozygous in the “test” individual and absent in the “background” individual; 2) Sequencing depth was required to be between 20× and 100× at all variable sites for all individuals; 3) The observed VAF had to be between 0.4 and 0.6 in the undiluted healthy cfDNA samples for the selected sites; 4) SNPs from repeat regions, as well as C->T variants at CpG sites were excluded; 5) SNPs present in multiple plasma samples were excluded. (Page 28, lines 608 - 616)

b. Subheadings inconsistently capitalized as sentence or initial case.

Only the first letter of the subheadings was capitalized.

c. Page 12: "Patients 2"

Changed to “Patient 2”

2. Inconsistent numerical annotations:

a. 20X and 100X some places, and 20x and 100x other places

X or x in 20X, 100X, 20x, 100x, 4.2×10^{-7} , 1×10^{-5} et.al were modified to ×

b. 5K, 10K, 20K some places, and 5000, 10000, 20000 other places

5000, 10000, 20000, 5K, 10K, 20K were changed to 5K, 10K, 20K, respectively.

Referee #2:

Li et al. report a WGS assay, AccuScan, for ctDNA-based detection of molecular residual disease (MRD). The assay utilizes rolling circle amplification (RCA) to control sequencing error rate, and couples it with a statistical framework for calling MRD status and estimating circulating tumor allele fraction (cTAF). The author report significantly reduced sequencing error rate from RCA compared to conventional WGS, and demonstrate high specificity and moderate to high sensitivity in MRD calls. The reported clinical applications are interesting and promising. The following methodology-related points should be clarified/addressed:

We thank the reviewer for all the insightful comments.

1. A key component of the assay is the use of RCA concatemers for error reduction. In Results, the authors state that a change from the reference that is consistent in all copies of concatemer is a presumed variant. In Methods, the authors define true variants as differences from the reference that were supported by at least two tandem copies within a read-pair. This inconsistency needs to be clarified.

We thank the reviewer for the suggestion. A change from the reference needs to be consistent in all copies of the concatemer, with a minimum of two copies sequenced, in order to be considered as a variant. We have made changes in the revised version so that the descriptions in the "Results" and "Material and methods" sections are consistent.

Revision in Results section:

A change from the reference that is consistent in all copies **and supported by at least two copies** is a presumed variant; **an** inconsistent **change** is likely due to a PCR or sequencing error and is removed. **(Page 7, lines 144 - 146)**

Revision in Materials and methods section:

Base calls that were consistently supported within a read-pair by **all** tandem copies **and supported by at least two copies** were considered **repeat confirmed**. **Difference from the reference calls that are repeat confirmed are called** as true variants; inconsistent differences **between tandem copies** were considered as random errors **and**

discarded. Repeat confirmed reference calls are counted towards molecule depth at the genomic location together with repeat confirmed variant calls. (Page 24, lines 531 - 536)

2. For patients without matching white blood cell (WBC) samples, the authors tested and utilized a WBC-free workflow, which distinguishes tumor-specific variants from germline and CHIP mutations by defining the former as variants found at single molecule level and the latter as those found at {greater than or equal to}2 molecules levels. It is unclear how this can be achieved as there is no mention of tracking unique molecules in Methods. It is also unclear whether the cutoff is supported by any data.

We thank the reviewer for the comments. The AccuScan method is tracking molecules using the boundaries of the molecules. We added additional information to clarify this in the Material and methods section. We established the cutoff based on the Poisson model (Appendix Figure S6). At 40× sequencing coverage, a germline variant with 50% or 100% allele frequency has $<4.32e-8$ probability of being detected with 0 or 1 variant molecules, indicating that majority, if not all, of the germline variants will be observed with ≥ 2 unique molecules and can be filtered out by these criteria. Similarly, CHIP variants can be filtered out if the CHIP is present at high frequency.

Revision in Results sections:

We used the start and end positions of a cfDNA molecule as the unique identifier to count the number of unique molecules (Materials and methods). (Page 9, lines 189 - 190)

When the tumor fraction is low, germline variants and somatic variants will be represented by different numbers of molecules in the post-treatment plasma samples. We modeled the distribution of the expected number of variant molecules observed under different cVAFs using a Poisson model (Appendix Figure S6). With 40-60× sequencing of cfDNA extracted from a low-tumor-burden (cVAF < 0.1%) plasma sample, germline SNVs (cVAF = 50% or 100%) are typically observed at ≥ 2 unique

cfDNA molecule levels, while tumor-specific variants (cVAF < 0.1%) are generally found at < 2 unique cfDNA molecule levels (Appendix Figure S6). Therefore, we may remove variants found with 2 or more molecules in the post-treatment plasmas from the tumor tissue sequencing results to obtain the list of tumor-specific mutations.

(Page 11, line 228 – 236)

Revision in Materials and methods section:

When WBC data was not available, we removed potential germline and high allele frequency CHIP variants by pairing the tumor sequencing data with sequencing data from the matching post-treatment plasma sample, which is expected to have a very low tumor burden. Tumor variants detected with more than one unique molecule in the post-treatment plasma sample are filtered out. **(Page 23, lines 515 - 519)**

Molecule boundaries are determined from neighboring alignments within the same read-pair and can be used for quantification. Molecule depth is calculated at each position and requires repeat confirmation for variant and wild-type calls. **(Page 24, lines 526 - 529)**

Appendix Figure S6. Distribution of probability and expected number of observed variants in Poisson model at different allele frequencies.

A Probability distribution of Poisson model using depth of 40 and circulating variant allele frequencies (cVAF) ranging from 0.1% to 50%. X-axis is the number of

observed variant molecules and y-axis is the probability to see this many variant molecules for a given variant for different cVAFs.

B Similar to A but showing the expected number of observed variants under the assumption of evaluating 5K variants as tumor-specific markers.

3. In the results related to Figure 3A-D, the authors state that "With 40-60× sequencing of a low-tumor-burden (cTAF < 0.1%) plasma sample, germline and CHIP mutations can be found at {greater than or equal to} 2 molecules level, while tumor-specific variants will generally be found at single molecule level." By removing mutations with {greater than or equal to}2 molecules, they show that their methodology is comparable to a workflow that had additional sequencing of white blood cells. Thus, they claim that this methodology can thus be used in "place of WBC" sequencing. However, per their determination above regarding "low-tumor burden", it could be inferred that this may not be the case for patient with higher-tumor burden (for example, the BRAF-mt patient that had cTAF >1% in the preceding result sections). Is this true? If not, please clarify in the results section how their determination that their approach can be used in place of WBC is dependent or not-dependent on tumor burden.

We thank the reviewer for the insightful comment. Appendix Figure S6 illustrates the percentage of single-molecule variants that are expected to be found in a sample at different cVAF based on Poisson distribution. For samples with cVAF < 15%, AccuScan would be able to detect a significant number of single-molecule variants and thus would call MRD positive correctly, albeit with an incomplete list of somatic variants. For samples with cVAF > 15%, majority of the tumor-specific variants will be present at 2 or more molecules. In these cases, AccuScan would report little or no tumor variant at single molecule level. Given that the tumor samples typically have thousands of cancer mutations, the observation of little or no tumor variant at single molecule level would suggest high tumor fraction in the plasma, and this hypothesis can be confirmed by other markers for MRD detection such as CNV or cfDNA fragment size, which are well suited for tumor detection at high tumor fraction. We

added text to address this case and clarify our position.

Revision in Results section:

2) In plasma samples with high tumor burden, a significant fraction of the tumor variants might be observed at ≥ 2 unique cfDNA molecule levels. However, as shown by Poisson model, at $cVAF = 15\%$, we will still be able to observe multiple variants with one molecule (Appendix Figure S6), suggesting that the effect on MRD sensitivity is likely to be minimal. The primary impact would be an underestimate of the tumor variant count and the observed $cVAF$ in plasma. 3) At very high $cVAF$ ($cVAF > 15\%$), the Poisson model predicts that the majority of cancer variants will be observed at the level of ≥ 2 unique cfDNA molecules. Consequently, comparison of the tumor tissue sequencing and plasma AccuScan data will remove most of the variants found in tissue, resulting in little or no tumor-specific variant as MRD markers. Given that tumor tissue samples typically harbor thousands of cancer mutations, the observation of minimal or no tumor variants at the single molecule level would suggest a high tumor fraction in the plasma. This hypothesis can be confirmed by other markers for MRD detection, such as copy number variation or cfDNA fragment size, which are well suited for cancer detection at high tumor fraction. (Pages 11-12, lines 236 - 249)

Appendix Figure S6. Distribution of probability and expected number of observed variants in Poisson model at different allele frequencies.

A Probability distribution of Poisson model using depth of 40 and allele frequencies ranging from 0.1% to 50%. X-axis is the number of observed variant molecules and y-axis is the probability to see this many variant molecules for a given variant.

B Similar to A but showing the expected number of observed variants under the assumption of evaluating 5K variants as tumor-specific markers.

4. The other key component of the assay is the statistical model for calling MRD status and estimating cTAF. The accuracy of cTAF estimation was assessed using a single BRAF mutation in a dilution experiment (Fig. 2E). Since a typical AccuScan assay involves tracking thousands of target mutations, the accuracy of cTAF estimation would be better assessed in such context, e.g. using the healthy sample mixtures described in the manuscript for assessing the performance of MRD calls, which involve 5,000 to 20,000 SNVs.

We thank the reviewer for the comments. We followed your suggestions and assessed the accuracy of cVAF estimation using the healthy sample mixtures at 2K, 5K, 10K and 20K variants. The data shows strong correlation of expected cVAF and observed cVAF and is now shown in the Appendix Figure S3 of the revised manuscript.

Appendix Figure S3. Observed circulating variant allele frequency point estimates in simulated subsamples of data from serial dilutions of three healthy

cfDNA. It was performed in 1000 simulations per concentration and number of variants. Expected circulating variant allele frequency (cVAF) ranged from 1×10^{-4} to 1×10^{-6} . Dots represent the mean of the point estimate, and the bars indicate 1 standard deviation.

5. How many WGS SNVs do circulating tumor DNA and normal samples usually differ at? An estimation should be provided when presenting results from the selectivity analysis employed throughout the study to allow a proper interpretation of the significance of the analyses. Indeed, in a later section their sequencing "identified 5,820 tumor-specific variants per patient (2148-265800)". Because as few as ~2K variants can be identified in patients, the authors should provide measures of sensitivity/specificity to at least down to 2K variants throughout their different sensitivity analyses (instead of down to 5K variants as currently employed).

We thank the reviewer for the comments and suggestions. In plasma samples, we saw very low number of variants in healthy and the number of variant molecules in cancer differs based on cVAF. The number of variant molecules observed in normal sample (cVAF = 0) and serial dilutions of a melanoma sample is now shown in Appendix Figure S4 of the revised manuscript.

Appendix Figure S4. Observed number of variant molecules of a melanoma cfDNA sample with expected circulating variant allele frequency from 1×10^{-3} to 2×10^{-6} and healthy controls. Experiments were performed with one test at 1×10^{-3} , two replicates at 1×10^{-4} , three replicates at 1×10^{-5} , 5×10^{-6} , 2×10^{-6} , and two replicates with the healthy control sample.

In tissue samples, we see ~2K-200K more variants in cancer tissues than in normal tissues. The tumor specific variant count is provided in the manuscript (Dataset EV4). We added simulation data for 2K variants in the revised Figure EV2A-B, and expanded our analytical sensitivity and specificity analysis to include the case of 2K variants as well in the revised Figure 2C-2D.

Revision in Results section:

The observed specificities were $\geq 99\%$ for 2K, 5K, 10K or 20K markers conditions (Fig 2C). The observed sensitivity at 2.5×10^{-5} cVAF and above level was greater than 99% for all conditions tested. At 10 PPM, corresponding to 1×10^{-5} cVAF, the average detection rate was 38% for 2K markers, 77% for 5K markers, 96% for 10K markers, and 100% for 20K markers (Fig 2D, Appendix Figure S3). **(Page 9, lines**

Revised Figure 2C-2D
C, D Observed analytical specificity (C) and sensitivity (D) with serial dilutions of healthy cell-free DNA (cfDNA) mixtures simulating cVAF from 1×10^{-4} to 1×10^{-6} . cfDNA from healthy donors ($n = 3$) was titrated independently into cfDNA from a different healthy “background” donor. Samples were processed by AccuScan with 10ng input, and tested for MRD by sampling 2K, 5K, 10K or 20K single nucleotide polymorphisms, 1000 times each, as tumor-specific markers. The error bars represent the likelihood-ratio based 95% confidence intervals.

6. Regarding the sensitivity analysis (Fig 2C), the authors state that "SNVs were selected to have a variant-type profile similar to CRC tumors". Similarly, in the methods, they state they "also constrained the subsamples to having the same distribution of mutation types as in a typical CRC tumor". What explicit criteria was used to determine "variant-type profile"?

We thank the reviewer for the suggestion. The variant-type profile is the percentage of different substitutions. We have included the variant-type profile of CRC (Appendix Figure S2) in the revised manuscript. The percentages of each variant types were obtained from whole genome sequencing of CRC tissue samples. We also added some text in the manuscript to describe how we have obtained the variant type profile for CRC.

Revision in Results section:

In each dilution, out of the over 100K SNVs distinguishing the test and background samples (Table EV1), we performed the test using randomly selected subsets of 2K, 5K, 10K, or 20K SNVs as markers, chosen to represent the variant-type profile typical of CRC tumors (Appendix Figure S2, Materials and methods). (Page 8, lines 174 - 177)

Revision in Discussion section:

For each plasma titration sample, we randomly subsampled 2K, 5K, 10K, or 20K variants constrained to having the same distribution of mutation types as in a typical CRC tumor. The variant profiles representative of CRC tumors was obtained by first collecting counts of variant for each variant type from tissue WGS data of each CRC patients, and then calculating the fraction of each variant type among the total somatic SNVs. The fraction of each variant type for each patient were also calculated and the distributions of frequencies were used to obtain the standard deviations. The CRC mutation profile was shown in Appendix Figure S2. (Page 28, lines 619- 626)

Appendix Figure S2. The representative profile of variant types for colorectal cancer patients. Point estimates are obtained as the fraction of each variant type in

the aggregation of variants from multiple colorectal cancer patients. Error bars are one standard deviation using the distribution of fractions in all patients.

Referee #3:

The methods section of the manuscript would benefit from more detail.

We thank the reviewer for the comments. We have added more details in the Material and methods section in the revised manuscript.

Referee #3 (Remarks for Author):

Review of the manuscript "Ultra-sensitive molecular residual disease detection through whole genome sequencing with single-read error correction" submitted by Xinxing Li et al for publication in EMBO Molecular Medicine

The authors present a novel approach for performing whole genome sequencing of plasma cell free DNA. The method includes several unique elements: 1) the input DNA is single stranded, 2) the DNA is circularized and amplified using rolling circle amplification, which generates concatemers of the input DNA, 3) from the single concatemer'ed DNA WGS libraries are generated and sequenced using a standard library kit and sequencer. 4) due to the replicative nature of the DNA it is possible to do error correction at single read level, and the authors report that their approach achieves an error rate of 4.2×10^{-7} which is ~2 orders of magnitude lower than other approaches.

The authors use the new approach for MRD detection in the setting of colorectal and esophageal cancer. They furthermore use it for assessing response to immunotherapy in Melanoma patients.

While the approach is novel, promising and the results are interesting I do have some concerns. It is important that the authors acknowledge that it is just a proof-of-concept study and that a thorough large and biased study, where ctDNA analysis is performed blinded to the outcome of the patients are needed to appropriately assess the performance.

We thank the reviewer for the constructive comments. We have revised the manuscript following the reviewer's suggestions.

Major comments

Comment 1:

The authors should clarify whether the investigators were blinded to the outcome of the patients when doing the ctDNA and bioinformatic analyses?

We thank the reviewer for the comment. During the early stage of the study, the outcome of the CRC and ESCC patients were not blinded. The monitoring analysis of the melanoma patients, which was conducted during the late stage of the study, was done with the patient outcome blinded to the analysts when doing the ctDNA and bioinformatics analysis. Our approach does not rely on machine learning that requires extensive data to be trained and could be susceptible to over-fitting, but rather on a simple statistical model that counts occurrences of true positive molecules and compares it against a sample specific, measured background error model. While this does not completely remove the risk of over-fitting, it drastically reduces that risk. We have added the information to the manuscript as you recommended.

Revision in Material and methods section:

The outcomes of the CRC and ESCC patients were not blinded, while the outcomes of the melanoma patients were blinded to the analysts when doing the ctDNA and bioinformatics analysis. (Pages 21 - 22, lines 472 - 474)

Comment 2: On page 8 the Authors write

"To further confirm the sample level specificity of AccuScan, we collected more than 1.3M tumor specific variants from 57 different cancer patients, including colorectal cancer (CRC), esophageal squamous cell carcinoma (ESCC) and melanoma, randomly sampled a subset of 5K, 10K or 20K equivalent variants for testing the MRD call in mismatched patient plasma samples. We did 2,000 random samplings of mismatched variants for each combination of variant count level and plasma sample. The average sample level specificity is computed as the fraction of MRD tests that are characterized with a negative MRD call. The observed values were similar to the nominal specificity, with 99.3%, 99.1%, and 98.9%, for 5K, 10K and 20K variant count levels, respectively. These results suggest that the AccuScan assay and analysis

have the intended performance for patient plasmas using tumor variants (Data not shown)." These results should be shown.

The authors should also clarify whether they have matched tumor, WBC and plasma on these 57 patients. I assume they do the testing in 57 plasma samples, but this is not clear. Are the plasma samples preoperative or postoperative? What is the sensitivity in the paired analysis (tumor/plasma) of the 57 patients? I Expect the fraction of positive MRD calls is higher when the mutations are matched to the plasma samples, than when they are mismatched. The authors should show this.

We thank the reviewer for the suggestions. We have added the data for the specificity analysis to the manuscript (Please see Appendix Figure S5 in the revised manuscript). A summary of all the patients and samples (including tumor, WBC and plasma) used in the analysis are now shown in Table EV9 of the revised manuscript. We did the test in 117 plasma samples from 57 cancer patients, including 43 pre-OP plasma samples, 52 post-OP plasma samples, and 22 plasma samples before or during immune therapy.

The sensitivity of tumor matched plasma samples can be calculated for each plasma with expected positive outcome. Pre-OP CRC and ESCC samples are detected at 100% sensitivity (43 out of 43 samples are MRD positive). Post-OP samples from patients with recurrence in CRC and ESCC have 81% Sensitivity (17 out of 21 plasma samples from recurrent patients are tested MRD positive). The sensitivity data for these samples are presented in Figure 4 of the revised manuscript. The fraction of positive MRD calls is significantly higher when the mutations are matched to the plasma samples than when they are mismatched.

Revision in Results section:

The observed average specificities are 99.6%, 99.3%, 99.1%, and 98.9%, for 2K, 5K, 10K and 20K equivalent variant count levels, respectively. These results suggest that the AccuScan assay and analysis achieve the intended performance for patient plasma samples using tumor variants (Appendix Figure S5). (Page 10, lines 211 - 214)

Appendix Figure S5. The observed specificity as a boxplot for each target number of variants (2K, 5K, 10K, and 20K) for 117 plasma samples from 57 patients. The average specificity across all plasma samples is above the nominal specificity of 99% set in the model.

Comment 3: Section: "Identification of tumor-specific variants using a white blood cell free workflow"

Central to the proposed tumor-informed approach is the determination of the patient specific catalog of tumor-specific variants. To be able to filter out non-tumor specific variants a normal DNA control is needed. To avoid sequencing matched normal DNA the authors suggest to use data from a postoperative plasma sample as normal control, instead. They motivate this, by claiming the the cTAF in such a sample will be very low (<0.1%) and with 40-60x sequencing of the plasma DNA, then all variants (but those occurring just once) are likely to be germline variants, which should be filtered from the tumor variant catalog. The authors claim they also filter CHIP variants by this approach. However, this not completely true. They only filter CHIP variants if their plasma allele frequency is very high >3% (2/60). CHIP variants with plasma allele frequencies below this level will not be filtered. This paper <https://doi.org/10.1038/s41591-019-0652-7> indicates that the vast majority of CHIP

variants (variants found in plasma and WBC, but not in the tumor) have plasma VAFs below 3% - most actually have plasma allele frequencies below 0.2%. The authors should revise the text to reflect this.

We thank the reviewer for the insightful comments. The reference paper mentioned is helpful. We have revised the text in the manuscript according to the reviewer's remarks.

Revision in Results section:

A tumor-informed MRD test uses tumor-specific variants as markers for tracking the disease. The most commonly used strategy for tumor-specific variant identification is through paired sequencing of the tumor tissue DNA and a matched normal DNA sample, for example, DNA from white blood cells (WBC) of the patient (Cancer Genome Atlas Research, 2011; Saunders et al, 2012). By comparing the sequences of matched tumor and normal DNA, paired sequencing can effectively remove germline variants, which can significantly outnumber somatic variants. However, this method requires additional sample processing and sequencing (Fig 3A). To simplify the MRD workflow, we explore the feasibility of omitting WBC sequencing and using the sequencing data from post-treatment plasma samples to pair with tumor tissue sequencing for germline variant removal and somatic variant identification (Fig 3B).

The challenge with this WBC-free approach is to distinguish between germline variants and potential cancer variants in post-treatment plasma samples. We addressed this issue by assuming that 1) When the tumor fraction is low, germline variants and somatic variants will be represented by different numbers of molecules in the post-treatment plasma samples. We modeled the distribution of the expected number of variant molecules observed under different cVAFs using a Poisson model (Appendix Figure S6). With 40-60 \times sequencing of cfDNA extracted from a low-tumor-burden (cVAF < 0.1%) plasma sample, germline SNVs (cVAF = 50% or 100%) are typically observed at ≥ 2 unique cfDNA molecule levels, while tumor-specific variants (cVAF < 0.1%) are generally found at < 2 unique cfDNA molecule levels (Appendix Figure S6). Therefore, we may remove variants found

with 2 or more molecules in the post-treatment plasmas from the tumor tissue sequencing results to obtain the list of tumor-specific mutations. 2) In plasma samples with high tumor burden, a significant fraction of the tumor variants might be observed at ≥ 2 unique cfDNA molecule levels. However, as shown by Poisson model, at cVAF = 15%, we will still be able to observe multiple variants with one molecule (Appendix Figure S6), suggesting that the effect on MRD sensitivity is likely to be minimal. The primary impact would be an underestimate of the tumor variant count and the observed cVAF in plasma. 3) At very high cVAF (cVAF > 15%), the Poisson model predicts that the majority of cancer variants will be observed at the level of ≥ 2 unique cfDNA molecules. Consequently, comparison of the tumor tissue sequencing and plasma AccuScan data will remove most of the variants found in tissue, resulting in little or no tumor-specific variant as MRD markers. Given that tumor tissue samples typically harbor thousands of cancer mutations, the observation of minimal or no tumor variants at the single molecule level would suggest a high tumor fraction in the plasma. This hypothesis can be confirmed by other markers for MRD detection, such as copy number variation or cfDNA fragment size, which are well suited for cancer detection at high tumor fraction.

To test the feasibility of this approach, we compared the performance of a tumor-WBC workflow with a WBC-free workflow using matched tumor tissue, WBC and plasma samples collected from 20 cancer patients (Table EV2 , Fig 3A-B). The number of tumor-specific variants and variant type profiles found by the two different workflows are shown in Figure 3C-D. Overall, the number of mutations identified by both methods were very similar, as were the variant-type profile of the mutations identified. AccuScan MRD analysis of the plasma samples (n=48, including 18 pre-treatment samples and 30 post-treatment samples) from these 20 patients showed identical MRD calls under either workflow (Fig 3E, Table EV2), and the cVAF values were strongly correlated ($R^2=0.99$, Fig 3F). These results suggest that post-treatment plasma can be used instead of WBC for the identification of tumor-specific variants.

In addition to germline variants, another potential complication in identifying

tumor-specific variants is the presence of clonal hematopoiesis of indeterminate potential (CHIP) variants. CHIP variants are somatic mutations present in hematopoietic cells. Study has shown that the variant allele frequencies (VAF) of CHIP mutations are strongly correlated in the matched cfDNA and WBC samples (Razavi et al, 2019). At 60× sequencing depth, the rule of removing variants with ≥ 2 unique cfDNA molecules will filter CHIP variants if their cVAF is $>3\%$ ($2/60$) in plasma. However, this rule will not remove CHIP variants with cVAFs below 3% in plasma. Meanwhile, most CHIP mutations are not observed in tumor tissue with high VAF ($> 5\%$). The probability of having a CHIP mutation found in the tumor tissue with high enough VAF ($> 5\%$) and simultaneously in cfDNA with one unique molecule is small. To prevent false positive calls caused by CHIP mutations, which can be mistaken as tumor-specific mutations present in post-treatment plasma under rare circumstances, sequencing the matching WBC sample when a plasma sample is marginally called MRD positive may be beneficial. **(Pages 10-13, lines 216 - 272)**

In the section the authors write "AccuScan MRD analysis of the plasma samples (n=48) from these 20 patients returned identical MRD calls under either workflow". The authors should explain which 20 patients and how they get to 48 plasma samples. Are the 48 samples all collected post-operatively or do the authors include preoperative plasma as well.

We thank the reviewer for the comments. The 20 patients included 6 CRC patients, 6 ESCC patients and 8 melanoma patients. All these 20 patients have matching post-treatment plasma samples sequenced relatively deep. The 48 plasma samples included 18 pre-treatment and 30 post-treatment samples from these 20 patients. Detailed information of the patients and associated plasma samples have been added to the manuscript in the extended Table EV2.

Revision in Results section:

AccuScan MRD analysis of the plasma samples (n=48, including 18 pre-treatment samples and 30 post-treatment samples) from these 20 patients showed identical MRD calls under either workflow (Fig 3E, Table EV2), and the cVAF values were

strongly correlated ($R^2=0.99$, Fig 3F). (Page 12, lines 255 - 258)

Comment 4: Tables S3-S5

The authors should provide sufficient clinical information to reproduce the results.

This includes adding outcome event information to Tables S3, S4, S5

For tables S3 and S4 this includes

- Event status (no event, recurrence, death)
- Time to event (or censoring)
- Adjuvant chemotherapy status

For Table S5 the information should include the response evaluations and the progression free survival

We have added detailed clinical information of the cancer patients in Tables S3-S5 (the newly Table EV3-EV5) following the reviewer's suggestions,

Revision in Expand tables section:

Tables S3 (Table EV3 in the revised manuscript) and Tables S4 (the Table EV4 in the revised manuscript): added the information of "Neoadjuvant therapy", "Adjuvant therapy", "Recurrence status", "Disease-free survival", "Death", "Overall survival".

Tables S5 (Table EV5 in the revised manuscript): added the information of "Plasma sampling time point", "Treatment", "Response evaluations", "Progress-free/Disease-free survival", "Death", "Overall survival".

Comment 5: Section MRD detection and prognostic value in surgical patients

The authors should specify the outcomes and results for the patients with an available landmark sample. It is confusing that they do not report specifically on the landmark.

They provide results for an analysis including all available postOP samples. This latter analysis may well be confounded/biased as it is clear from the swimmer plots in Figure 3 that the non-landmark samples were not systematically collected. For the CRC recurrence patients non-landmark samples are only available when the landmark was falsely negative or not available.

The authors should report the landmark results separately for a unbiased and clearer

presentation.

We thank the reviewer for the suggestion. The Kaplan-Meier curve for landmark is shown in Figure 4D. In addition, we have added the specificity and accuracy of the landmark in CRC cohort. We added columns "Post-OP plasma sampling (landmark)" and "Post-OP plasma sampling time (months since surgery)" in Table EV3 (previous table S3) and marked the ctDNA test results of each time point.

Revision in Results section:

Taken together, these results suggest 90% (95% CI: 55.5%-99.8%) sensitivity, 100% (95% CI: 81.5%-100%) specificity and 96.3% (95% CI: 81.0%-99.9%) accuracy at landmark (within 6 weeks after surgery), 100% (95% CI: 71.5%-100%) sensitivity, 100% (95% CI: 88.1%-100%) specificity, and 100% (95% CI: 91.2%-100%) accuracy with longitudinal monitoring for predicting CRC recurrence. **(Page 14, lines 299 - 303)**

Comment 6: Discussion section

The authors provide present data and results in the discussion section (Figure S5). This is inappropriate. The authors should present the results of Fig S5 in the results section.

We thank the reviewer for the comment. We have moved Fig S5 to the result section and reordered it as Figure EV1.

Revision in Results section:

AccuScan with concatemer error correction had an average error rate of 4.2×10^{-7} , which is ~2,000-fold lower than the unfiltered WGS data, and ~67-fold reduction when compared to the read-pair corrected WGS (Fig 1B, Appendix Figure S1). The AccuScan error rate can be further reduced to 3.1×10^{-7} when sequenced using the single-end 300 (SE300) base sequencing (Fig EV1). **(Pages 7 - 8, lines 150 - 154)**

Comment 7: Discussion section

The reported results can only be seen as proof-of-principle. Nowhere in the manuscript is there a description of the process and criteria for including patients. The

patients and samples seem to have been highly selected. For instance, the CRC cohort, the recurrence cases are seemingly much younger than the non-recurrence patients (Table S3). Furthermore, from Figure 3, one can see the collection time-points of the landmark samples are also biased. For the recurrence patients the landmark samples were collected at between month 1 and 2, while it for the majority of the non-recurrence patients were collected within a week or two of surgery. This difference could indicate the samples were not collected as part of the same study and maybe not using the same procedure either. We cannot tell as the authors do not provide sufficient detail.

While this is suboptimal, I don't see it as a major problem for a proof of principle paper. Though, one should be careful not to compare the sensitivity and specificity obtained from such biased cohorts with studies reporting on consecutive non-selected cohorts where the samples were collected using the standardized procedures.

Therefore I strongly recommend the authors to reformulate or remove the following statement in the discussion "Considering an average sensitivity of 45%-56% of landmark sensitivity for CRC with current commercial offerings, AccuScan picked up as much as twice the number of patients who may benefit from adjuvant therapy without increasing over treatment."

For the comparison to make sense, the AccuScan performances need to be confirmed in independent, large, and consecutively collected patient cohorts. Importantly, the AccuScan investigators should perform these analyses blinded to the patient outcome.

We thank the reviewer for the comments. We agree with the reviewer that this is only a small proof-of-principle study with retrospective samples. Validation of AccuScan with an independent, large, and consecutively collected patient cohort with blinded patient outcome is important for confirming the clinical performance. We have reformulated the corresponding part of discussion in the revised manuscript following the reviewer's suggestion.

Revision in Discussion section:

The sensitivity and specificity requirements for an MRD test depend on the specific applications. For example, National Comprehensive Cancer Network guideline

recommends starting adjuvant chemotherapy for high-risk CRC patients no later than 6-8 weeks after surgery (Benson et al, 2021). The landmark timepoint, which is before week six after surgery, is key for clinicians to decide between escalation or de-escalation of therapy. Current commercial offerings showed an average sensitivity of 41%-56% at landmark for CRC (Parikh et al, 2021; Reinert et al, 2019; Tarazona et al, 2019). Faster TAT and higher sensitivity at landmark are needed to give oncologists the time and confidence to de-escalate treatment while ensuring most, if not all, high risk patients receive needed adjuvant therapy following their surgery. On the other hand, during longitudinal monitoring for patients who have shown partial response or complete response, TAT is not as critical as the landmark time point, but high specificity is crucial for avoiding anxiety, unnecessary exposure to toxic side effects and financial burden associated with the treatments. In this report, AccuScan demonstrated analytical sensitivity at PPM level with contrived samples and achieved a landmark sensitivity of 90% for predicting relapse after surgery with 100% specificity in retrospective CRC patient samples. Taken together, these results indicate that AccuScan test has the potential to achieve very high sensitivity and specificity for MRD detection. It is important to note that the data presented here are from proof-of-principle studies. Large, blinded clinical studies with consecutively collected patient cohorts and sufficient follow-up (i.e., ≥ 3 years) are needed to confirm and validate the clinical performance of AccuScan. (Pages 19-20, lines 417 - 435)

Comment 8: Experimental Design

The authors should explain where and when the patients were recruited. They should also explain and justify the apparently biased variation in the landmark sampling time-points of the CRC cohort.

We thank reviewer for the comment. We have added more details in the revised version. In this proof-of-principle study, we used the remaining plasma samples in sample banks that had a sample volume of 1 ml or more. The CRC patient samples were collected from two sample banks of two different hospitals. Shanghai

Changzheng Hospital collected post-surgery blood samples within one week after surgery, while Tongji Hospital collected blood samples after surgery at different time points. The different sample collection procedures together with small sample size may have led to variations at landmark time-points within the CRC cohort. Among all the non-relapse patients, there are 12 plasma samples collected within 1 week after surgery, 11 plasma samples collected more than one week post-surgery (Table EV3), and the observed specificity in all the non-relapse patients is 100%. Sampling time did not show significant impact on specificity in this study. Nevertheless, we agree with the reviewer that it is important to confirm and validate the AccuScan performance with a large, uniformly, and consecutively collected patient cohorts.

Revision in Materials and methods section

Participants were recruited in four hospitals. The CRC patients' samples were collected from Shanghai Changzheng Hospital (IRB: CZ2017-251) and Tongji Hospital (IRB: K-KYSB-2021-005) and from July 2017 to February 2023. The ESCC patients' samples were collected from Shanghai Changhai Hospital (IRB: CHEC2020-021) from June 2020 to December 2021. The melanoma patients' samples were collected from Yale School of Medicine (IRB: 0609001869) from March 2019 to February 2023. A total of 58 subjects including 32 CRC patients, 17 ESCC patients and 9 melanoma patients were included in this report. Inclusion and exclusion criteria were listed in Table EV8. (Page 21, lines 460 - 467)

Comment 9: Section Data and materials availability.

The authors claim that the cfDNA library preparation and variant calling using concatemers have been previously reported in detail (Refs 38 and 45). However, these references are insufficient. In ref38 - the mat and met sections refers back to another paper for description of how CLAMP-seq is performed. In ref 45, the methods used are only described in the supplementary materials and only very briefly.

I highly recommend the authors to provide sufficiently detailed method information in the present manuscript, to allow the readers to reproduce the results.

For example in the section "AccuScan error correction algorithm and error rate measurement" the authors write "Variant calls using concatemer has been previously described in (45). Summarizing, reads were aligned to the reference genome (hg38) and high read quality single nucleotide substitutions different to the reference are further investigated for repeat confirmation."

This is insufficient, they need to specify the definition of "high read quality single nucleotide substitutions"

We thank the reviewer for the suggestions. We significantly expanded the description of the methods used in this manuscript. We expanded on the specific thresholds (Q-score >24) and further described how variant and wild-type molecules are quantified using molecule boundaries and repeat-confirmation requirements. We further expanded descriptions of the error rate estimation and added more information to our MRD algorithm.

Revision in Materials and methods section:

Variant calls using concatemer has been previously described in (Xu et al, 2017). Summarizing, paired-end fastq files were processed using a nextflow pipeline (Di Tommaso et al, 2017) and Amazon Web Services (AWS) cloud computing to obtain variant calls and molecule depth estimates at each position, as well as an error rate estimate for the given plasma sample. First, reads were aligned to the reference genome (hg38) using bwa mem (Li, 2013). Molecule boundaries are determined from neighboring alignments within the same read-pair and can be used for quantification. Molecule depth is calculated at each position and requires repeat confirmation for variant and wild-type calls.

Each position within the molecule boundaries is evaluated for repeat confirmation, where only positions in the read with high read quality (Q-score>24) are considered. Base calls that were consistently supported within a read-pair by all tandem copies and supported by at least two copies were considered repeat confirmed. Difference from the reference calls that are repeat confirmed are called as true variants; inconsistent differences between tandem copies were considered as random errors and discarded. Repeat confirmed reference calls are counted towards molecule depth at

the genomic location together with repeat confirmed variant calls. (Page 24, lines 522 - 536)

AccuScan MRD algorithm

The AccuScan MRD algorithm is applied on a plasma sample with error corrected variant calls. It further requires a list of target SNVs and evaluates whether the observed number of variants in the plasma overlapping this list of markers exceeds the expectation from a background error model. The error model is measured for each plasma as described above. For each target variant position, the algorithm collects the repeat confirmed molecule depth and distinguishes between reference and variant counts. Unless the number of detected variants is exceedingly high, and thus indicating a high tumor burden sample, one can assume that many or most of the detected somatic variants are expected to occur only with a single molecule (Appendix Figure S6), therefore variants with more than one cfDNA molecule detected in the plasma are excluded for the MRD call in a low tumor burden sample. In addition to a positive/negative MRD call, the relative amount of tumor DNA is estimated using a probabilistic model, where the number of tumor variant molecules was modelled as a random variable that depends on the sequencing depth, error rates, and cVAF. (Page 25, lines 551 - 563)

Comment 10: Section Data and materials availability.

Related to the previous comment. The authors should make available the code used in the paper - this includes the error rate assessment and MRD algorithm codes. This is central for the readers and reviewers to be able to assess the appropriateness and quality of the used procedures. This is also central for others to be able to confirm and hopefully independently reproduce the results.

I understand that the raw sequencing data cannot be shared, however the variant lists should not be sensitive and should be made available for all patients/samples (before and after filtering).

The code should make it possible for readers of the paper to reproduce the results from the variant call files.

A table should be provided with all patients and samples, including an indication of what samples were included in which analyses (preoperative, landmark, longitudinal). We thank the reviewer for the comments. We have uploaded codes for error rate calculation , MRD status and cVAF estimation with respective input data on Zenodo and provided the URL in the revised version. We also uploaded variant lists for all individuals on Zenodo as well. We have added Table EV9 in the revised manuscript to include all the subjects and samples used in different data analysis.

Revision in Expand tables section: Table EV9

Comment 11: Section AccuScan error correction algorithm and error rate

How dependent are the estimated error rates of the 200Mb selected positions? The authors should repeat the selection of the 200 Mb positions and report the error rates with variance estimates.

The authors should motivate why the C to T variants at CpG sites were excluded.

The authors write "Within a given plasma sample, we removed variant positions where two or more variant molecules" However, re-occurring errors are still errors and should be included when reporting the error estimates of the AccuScan seq approach.

If the authors in practice always filter out variants with 2 or more cfDNA molecules detected in plasma, then they should motivate this, and explain why re-occurring errors should be removed. Then it would also be justified to also show the error rates based on this procedure.

We thank the reviewer for the comments. Following your suggestion, we have repeated the selection of the 200 Mb positions through 20 iterations in three normal samples and reported the error rates with variance estimates (Please see Appendix Figure S1 in the revised manuscript). The overall average error rate of three individuals was $4.26e-07$. The average error rate of the 20 iterations was $4.31e-07$, $4.55e-07$, $3.94e-07$ for healthy individual #1, #2, #3, respectively. This variation is small and comparable to the average error rate reported in figure 1B ($4.21e-07$) using the default 200Mb positions. We have now added Appendix Figure S1 in the revised

manuscript to illustrate the error for the different sets of positions and added text to describe the results.

The C to T variants at CpG sites were excluded for MRD calling and error rate analysis for the following reasons. Studies have shown that the cytosines in the CpG dinucleotide are hotspots for C to T substitution due to spontaneous deamination of methyl-C. (Coulondre et al, 1978; Hwang & Green, 2004). As a result, the observed C to T variants at CpG sites could be attributed to deamination of C on one strand, rather than mutations on both strands. In addition, the CpG dinucleotides comprise < 1% of the human genome (Babenko et al, 2017). Therefore, excluding C to T variants at CpG sites can significantly reduce the noise without sacrificing too much of the sensitivity of the test.

In practice, our MRD algorithm removes variants detected with 2 or more unique molecules as default. We have significantly expanded our methods section describing our algorithms, where we also highlight and motivate the exclusion of variants in the MRD model with 2 or more variant molecules. In summary, low cVAF samples have tumor-variants present almost exclusively at single-molecule level. To avoid artifact, we exclude multi-molecule variants in both error estimate and MRD call. Higher cVAF (e.g. >1%) samples have signal that is multiple orders of magnitude above the error and do not require this restriction.

Revision in Materials and methods section:

For computational reasons, error rates were calculated for a set of randomly selected ~200Mb positions of the genome. To ensure that this doesn't bias the estimate, we performed tests 20 times using different sets of 200Mb positions. In this test the overall error rate ranges from 4.0×10^{-7} to 4.5×10^{-7} with an average of 4.3×10^{-7} . This variation is small and would have minimal impact on our MRD calls (Appendix Figure S1). Error rates were calculated for every possible SNV in the selected loci, except for C to T variants at CpG sites, which were excluded for overall error rate calculation. To remove most germline variants in the error estimate of plasma samples, we filter variants using the gnomad (3.1) database (Karczewski et al, 2020). Additionally, variants that would not be considered for MRD due to being in variant

positions where two or more variant molecules were observed, are also excluded in this calculation. The remaining variants are sorted into variant types and error rates are calculated for each variant type as the sum of all observed variant molecules divided by the total sum of molecules interrogated for each variant type, i.e. only variants that pass our previous filter criteria are contributing towards the numerator and denominator in this equation. **(Pages 24 - 25, lines 537 - 550)**

AccuScan MRD algorithm

The AccuScan MRD algorithm is applied on a plasma sample with error corrected variant calls. It further requires a list of target SNVs and evaluates whether the observed number of variants in the plasma overlapping this list of markers exceeds the expectation from a background error model. The error model is measured for each plasma as described above. For each target variant position, the algorithm collects the repeat confirmed molecule depth and distinguishes between reference and variant counts. Unless the number of detected variants is exceedingly high, and thus indicating a high tumor burden sample, one can assume that many or most of the detected somatic variants are expected to occur only with a single molecule (Appendix Figure S6), therefore variants with more than one cfDNA molecule detected in the plasma are excluded for the MRD call in a low tumor burden sample. In addition to a positive/negative MRD call, the relative amount of tumor DNA is estimated using a probabilistic model, where the number of tumor variant molecules was modelled as a random variable that depends on the sequencing depth, error rates, and cVAF. **(Page 25, lines 551 - 563)**

Appendix Figure S1. The error rates of the 200Mb randomly selected positions in three healthy individuals through 20 iterations.

Comment 12: Section AccuScan MRD algorithm

The text reads "The number of tumor base calls was modelled as a random variable that depends on the sequencing depth, error rates, and tumor allele frequency"

With the reference to tumor allele frequency do the authors refer to the allele frequency observed in the tumor or to the circulating tumor allele frequency (cTAF)

This should be explained.

The authors mention that the cTAF estimated by maximizing the likelihood of the observations with respect to the variant allele frequency and the error rate. If they refer to the variant allele frequency in circulation it would be appropriate to label it cVAF.

We thank the reviewer for the comment. In the text "The number of tumor base calls was modelled as a random variable that depends on the sequencing depth, error rates, and tumor allele frequency", the tumor allele frequency refers to the circulating variant allele frequency. We have modified the sentence in the manuscript to avoid confusion.

We used the terminology “cTAF” based on the references cited below, which describes cTAF as the value that estimated the “fraction of mutant containing reads contributed from the tumor to the cfDNA sample”, which is different from “the fraction of tumor genomes in cfDNA as cTF”. Having said that, we agree with the reviewer that “cVAF” would be more clearly different from cTF (tumor fraction) and less confusion than cTAF. We therefore changed all the cTAF labels to cVAF in the revised manuscript as suggested by the reviewer.

Definition and description of cTAF from (Jamshidi et al, 2022)

Jamshidi et al reported “For each sample, somatic tumor variants were identified from available tumor-tissue biopsy and matched WBC sequencing. For those variants, cTAF was estimated based on the fraction of mutant containing reads contributed from the tumor to the cfDNA sample using the targeted small variants panel (TS)...We refer to the fraction of tumor genomes in cfDNA as cTF, which is the target of the inference.....

Mutant allele counts were approximated as a Poisson distribution where the rate was controlled by the local depth of sequencing, cfDNA allele frequency, and tumor biopsy allele frequency. Individual variants were treated as independent for purposes of aggregating the total likelihood. The posterior density was computed using a uniform prior. We define the estimated cTF to be the value at the median of the posterior density. Approximate credible intervals were obtained based on the same density. cTAF was then computed by multiplying the estimated cTF by the median tumor biopsy mutation allele frequency per sample to obtain an expected tumor mutant allele frequency in cfDNA. This provided a simple, robust method for estimating cTAF across samples with multiple orders of magnitude difference in value.”

Definition and description of cTAF from (Bredno et al, 2022)

It was reported that "cTAF is a modeled value that estimates the expected fraction of cfDNA in the circulation that originates from a tumor and contains a tumor-specific allele. Observations of allele frequencies at multiple loci of the tumor genome are

combined into a single cTAF estimate. cTAF is therefore an attractive metric to evaluate the performance of an MCED test as it estimates the expected abundance of tumor-specific alleles in a sample. Consequently, the number of tumor-specific features for MCED tests is calculated as the total cfDNA fragments multiplied by cTAF. For example, with an MCED test that scans 105 regions to a depth of 100× per region, a sample with a cTAF of 10⁻⁴ would have an expected yield of 1000 tumor-specific alleles. As ctDNA has a short half-life (from 16 minutes to 2.5 hours), tests based on ctDNA capture the status and activity of a tumor close to the time of a blood draw. Figure 1 visualizes the underlying biology that drives ctDNA levels in an individual with cancer and how the resulting set of frequencies of alleles at variant loci drives the estimation of cTAF.

Revision in Materials and methods section:

In addition to a positive/negative MRD call, the relative amount of tumor DNA is estimated using a probabilistic model, where the number of tumor variant molecules was modelled as a random variable that depends on the sequencing depth, error rates, and cVAF. (Pages 25 - 26, lines 561 - 563)

We estimate the cVAF by maximizing the likelihood with respect to the variant allele frequency (*VAF*), with the constraint of ($VAF \geq 0$) and the observed error rate of the sample. (Page 26, lines 564 - 565)

Comment 13: Section AccuScan MRD algorithm

In the error rate section, the authors explained that they filter out all variants for which 2 or more reads are available when estimating the cfDNA error rate.

However, it is unclear to me if this is also done, when counting how many cfDNA fragments (observations) of a given variant type are in patient plasma samples (the Cv's).

This should be specified.

We thank the reviewer for the comment. We added some text to clarify that we treat both error rate calculation and MRD calling the same with respect to our filters. During error rate calculation, only variants that pass our filters contribute to the

numerator and denominator. For example, variants with 2 or more molecules found are not considered as error but are also not adding to the total depth for their variant type.

Revision in Materials and methods section:

Error rates were calculated for every possible SNV in the selected loci, except for C to T variants at CpG sites, which were excluded for overall error rate calculation. To remove most germline variants in the error estimate of plasma samples, we filter variants using the gnomad (3.1) database (Karczewski et al, 2020). Additionally, variants that would not be considered for MRD due to being in variant positions where two or more variant molecules were observed, are also excluded in this calculation. The remaining variants are sorted into variant types and error rates are calculated for each variant type as the sum of all observed variant molecules divided by the total sum of molecules interrogated for each variant type, i.e. only variants that pass our previous filter criteria are contributing towards the numerator and denominator in this equation. (Page 25, lines 541 - 550)

Comment 14: procedure for calling plasma samples ctDNA positive/negative

The authors should motivate the cut-off they have defined for calling a sample ctDNA positive or not. They write that they reject the null hypothesis when the cumulative distribution of a χ^2 distribution with 1 degree of freedom evaluated at the difference in log-likelihoods is greater 0.98"

What is the 0.98?

Moreover, the authors should explain when this cut-off was defined. Was it done before the study was initiated and blinded to patient outcome. Was it changed during the study to get the optimal separation between recurrence and non-recurrence patients?

I will recommend the authors to acknowledge in the manuscript, that to get a confident assessment of AccuScan performance at the landmark time point for colorectal and esophageal cancer, then the approach should be locked (the lab approach, the presented statistical procedure and the threshold for calling the samples)

and applied to an independent unbiased patient cohort (or cohorts) with at least 3-years of follow-up.

We thank the reviewer for the comment. We understand that the two values of 0.98 and 0.99 may lead to confusion. The target Specificity is set as 0.99. To achieve this, we evaluate the chi-square distribution with 1 degree of freedom at twice the difference in log-likelihoods. A threshold of 0.98 then corresponds to a P-value <0.01 following the methodology as described in the statistic reference in (Chernoff, 1954). We added some additional text to clarify the meaning of the values. We have added the content in the discussion section as you recommended.

The cutoff was defined to achieve 99% specificity. During development we evaluated thresholds of 98% and 99% specificity and didn't see a difference in sensitivity of our clinical samples and only one plasma sample as FP (matching the expectation from 98% specificity). After fixing the threshold we evaluated the observed specificity further using empirical data (see AccuScan clinical specificity in Materials and Methods).

We agree with the reviewer that this is only a small proof-of-principle study with retrospective samples. Validation of AccuScan using a locked approach with an independent, large, and consecutively collected patient cohort with blinded patient outcome is important for confirming the clinical performance. We have reformulated the corresponding part of discussion in the revised manuscript following the reviewer's suggestion.

Revision in Materials and methods section:

To test this hypothesis, we first consider that the null hypothesis corresponds to an extreme of the range of possible values for the parameter ($VAF \geq 0$ always, and for the null hypothesis $H_0: VAF = 0$), then find the difference in log-likelihoods at the point estimate of VAF and at 0, and reject the null hypothesis when the cumulative distribution of a χ^2 distribution with 1 degree of freedom evaluated at twice the difference in log-likelihoods is greater than 0.98. This corresponds to a P-value < 0.01 for the mixture distribution (delta and χ^2) that corresponds to testing as null an extreme of the possible values of the parameter (VAF

= 0), and therefore a specificity of 0.99 (99%) for the likelihood ratio test (Chernoff, 1954). 95% confidence intervals were obtained in the standard way based on the likelihood ratio: The maximum likelihood over values of VAF is obtained and its logarithm is computed. The non-negative values of VAF such that twice the difference of log-likelihood with the maximum is less than 3.841 are included in the confidence bounds for cVAF (two-tailed confidence intervals for positive samples and one-tailed upper bound for negative samples). This is derived from the asymptotic distribution of this statistic χ^2 . We note that the confidence interval does not overlap 0 in a sample that is called positive (VAF > 0). **(Pages 26-27, lines 582 - 596)**

Revision in Discussion section:

In this report, AccuScan demonstrated analytical sensitivity at PPM level with contrived samples and achieved a landmark sensitivity of 90% for predicting relapse after surgery with 100% specificity in retrospective CRC patient samples. Taken together, these results indicate that AccuScan test has the potential to achieve very high sensitivity and specificity for MRD detection. It is important to note that the data presented here are from proof-of-principle studies. Large, blinded clinical studies with consecutively collected patient cohorts and sufficient follow-up (i.e., ≥ 3 years) are needed to confirm and validate the clinical performance of AccuScan. **(Page 20, lines 428 - 435)**

Comment 15: cTAF 95% confidence bounds

The authors write "We obtained 95% confidence bounds for our cTAF estimates: two-tailed confidence intervals for positive samples and one-tailed upper bound for negative samples.

The approach used for this should be explained in detail.

We thank the reviewer for the suggestion. We have expanded the explanation.

Revision in Materials and methods section:

95% confidence intervals were obtained in the standard way based on the likelihood ratio: The maximum likelihood over values of VAF is obtained and its logarithm is

computed. The non-negative values of VAF such that twice the difference of log-likelihood with the maximum is less than 3.841 are included in the confidence bounds for cVAF (two-tailed confidence intervals for positive samples and one-tailed upper bound for negative samples). This is derived from the asymptotic distribution of this statistic χ^2 . We note that the confidence interval does not overlap 0 in a sample that is called positive (VAF > 0). (Page 27, lines 590 - 596)

Comment 16: the confidence in the ctDNA calls and cTAF estimates

The confidence in the ctDNA calls and cTAF estimates made by the AccuScan approach will be much improved if the authors run an well-established ctDNA detection strategy (e.g. digital PCR) on a subset of patients/samples (paired sample aliquot) and show the agreement between the two approaches in ctDNA calls and estimated cTAFs.

We thank the reviewer for the comment. We agree with the reviewer that running the same sample with a well-established ctDNA detection strategy such as ddPCR would improve the confidence in the ctDNA calls and cTAF estimates. Unfortunately, our experiments are constrained by the availability of patient samples. We only had no more than 4mL of plasmas for all the experiments reported in this study. In addition, the limit of detection of ddPCR at single locus is $\geq 0.1\%$, which is not sufficient to measure ctDNA at very low tumor fractions ($< 0.01\%$). Due to inadequate cfDNA samples and the technical limitations of ddPCR, we are sorry that we are currently unable to perform such an experiment.

Minor comments:

1) In the abstract the authors should specify

- a) the definition of the "landmark",
- b) the performance of the approach for detection MRD in Esophageal cancer
- c) what is meant by predictive value for immunotherapy monitoring

We thank the reviewer for the comments. We have added the content in the abstract as you suggested.

Revision in Abstract:

AccuScan showed 90% landmark sensitivity (within 6 weeks after surgery) and 100% specificity for predicting relapse in colorectal cancer. It also showed 67% sensitivity and 100% specificity with esophageal cancer using samples collected within one week after surgery. When AccuScan was applied to monitor immunotherapy in melanoma patients, the circulating tumor DNA (ctDNA) levels and dynamic profiles were consistent with clinical outcomes. (Page 3, lines 48 - 53)

2) Fig S2: There are two panels in A), B) and C). It is not explained what they reflect. We thank the reviewer for the comment. The two panels indicate the results of two independent replicates of each sample for ddPCR tests.

Revision in legend of Figure EV2:

C ddPCR of the diluted melanoma cancer cfDNA sample in the healthy plasma background at an expected circulating variant allele frequency of 0.1%. Each shows the results of two independent replicates of ddPCR tests. (Page 41, lines 1000 - 1002)

3) The citation provided for Ref 38 is incomplete

We thank the reviewer for the comment. We have added the volume and pages of Ref 38 and re-formatted the references according to the requirements of EMBO Molecular Medicine.

Revision in Reference:

Wang F, Li X, Li M, Liu W, Lu L, Li Y, Chen X, Yang S, Liu T, Cheng W et al (2024) Ultra-short cell-free DNA fragments enhance cancer early detection in a multi-analyte blood test combining mutation, protein and fragmentomics. Clin Chem Lab Med 62: 168-177

4) Section "AccuScan MRD algorithm". The authors write "We estimate the cTAF by maximizing the likelihood of the observations with respect to the variant allele frequency (VAF , $VAF \{ \text{greater than or equal to} \} 0$) and the error rate". What is meant by (VAF , $VAF \{ \text{greater than or equal to} \} 0$) ?

We thank the reviewer for the comment. What we meant here is VAF with the

constraint that $VAF \geq 0$. We have revised the manuscript to clarify the method.

Revision in Materials and methods section:

We estimate the cVAF by maximizing the likelihood with respect to the variant allele frequency (VAF), with the constraint of ($VAF \geq 0$) and the observed error rate of the sample. (Page 26, lines 564 - 565)

28th Jun 2024

Dear Dr. Weng,

Thank you for submitting your revised study, and please accept my apologies for the delay in getting back to you, as one referee needed more time to submit his/her report. We have now received feedback from the three initial referees. As you will see below, they are overall satisfied with the revisions, and I will therefore be able to accept your manuscript once the following points will be addressed:

1/ Referees' comments: please address the remaining concerns from referee #1 and carefully proof-read the manuscript for grammar and typos.

2/ Manuscript text:

- Please accept all changes, and only keep in track changes mode any new modification.

- Please note that institutional email addresses are missing for Hai Jin and Zhiqian Hu. An ORCID identifier is missing for Hai Jin.

- We note that you currently have together with you, a total of 4 co-corresponding authors. Is that correct? Do you confirm equal contribution of these 4 people, able to take full responsibility for the paper and its content? While there is no limit per se to the number of co-corresponding authors, 3 is rare, 4 even more so, and may not reflect as intended to the community.

- Methods:

o Patient samples: please include the full statement that the experiments conformed to the principles set out in the WMA Declaration of Helsinki and the Department of Health and Human Services Belmont Report.

o We encourage the inclusion of a Reagents and Tools Table. A downloadable template (.docx) for the Reagents and Tools Table can be found in our author guidelines:

<https://www.embopress.org/page/journal/17574684/authorguide#structuredmethods>

- Data availability:

o According to the journal's data policy, if practically possible and compatible with the individual consent agreement, we have to make sure that the authors deposit the human clinical datasets to public databases at the time of publication. Have you considered the following database: <https://ngdc.cncb.ac.cn/gvm/home?>

o Please remove "The datasets used and/or analyzed during the current study are available in the main text or the Expand tables or Datasets."

o Please make sure the deposited data are public before acceptance of the manuscript.

- References: only 10 authors should be listed before et al.

3/ Figures:

- Table EV9 should be made Dataset EV1.

- The table legends should be removed from the main manuscript text.

- Please address the following queries from our data editors:

1. Please note that the box plots need to be defined in terms of minima, maxima, centre, bounds of box and whiskers, and percentile in the legends of figures 1b; 3e; 4a-b; EV 1.

2. Please note that information related to n is missing in the legends of figures 4a-b.

3. Although 'n' is provided, please describe the nature of entity for 'n' in the legend of figure 3e.

4. Please note that the error bars are not defined in the legend of figure 2d.

4/ Checklist:

- Please fill in the full 'Experimental study design and statistics'

5/ I slightly edited your Paper Explained. Please let me know if you agree with the following or amend as you see fit:

Problem

Timely and accurate measurement of molecular residual disease (MRD) is critical for recurrence risk assessment and treatment decisions. Whole genome sequencing (WGS) for circulating tumor DNA (ctDNA) detection is an emerging field, enabled by the recent drop in sequencing cost. While WGS has shown feasibility for MRD detection, its wide use has been hampered by low sensitivity and specificity due to the lack of an effective error correction mechanism at the whole genome scale.

Results

In this study, we introduce AccuScan, a highly efficient cell-free DNA (cfDNA) WGS technology that significantly reduces the error rate by orders of magnitude. AccuScan demonstrates limit of detection at the parts per million (PPM) range in analytical studies and achieves high accuracy in predicting relapse in post-surgery patients with colorectal or esophageal cancers. In melanoma patients undergoing immunotherapy, AccuScan-measured ctDNA dynamics shows strong correlations with clinical outcomes. When combined with tumor sequencing, AccuScan offers a white blood cell-free workflow that greatly simplifies the process of tumor-informed molecular residual disease (MRD) detection.

Impact

AccuScan is a novel cfDNA WGS technology featuring genome-wide error suppression at single-read level. It enables a white blood cell-free approach for identifying tumor-specific variants and detecting tumor-informed MRD. With its ultralow limit of detection, minimal sample requirement, rapid turnaround time and comprehensive molecular information, AccuScan holds great potential for a wide range of research and clinical applications.

6/ Synopsis:

- Thank you for providing a synopsis image. Please resize it as a png/tiff/jpeg file 550px wide x 300-600 px high, and make sure that the text remains legible.
- I introduced minor changes to your text, please let me know if you agree with the following or amend as you see fit:

"A novel approach, named AccuScan, was developed for molecular residual disease (MRD) detection and immunotherapy monitoring. This technology uses whole genome sequencing (WGS) with single-read error correction for circulating tumor DNA (ctDNA) analysis.

- AccuScan reduced WGS error rate to less than 5×10^{-7} , enabling an ultralow limit of detection for circulating tumor variant allele frequency down to the parts-per-million (PPM) range.
- When applied to MRD detection, AccuScan achieved 90% landmark sensitivity and 100% specificity in post-surgical colorectal cancer patients.
- AccuScan showed high sensitivity and specificity for MRD detection in esophageal cancer and for immunotherapy monitoring in melanoma patients.
- AccuScan established a simple, white blood cell-free workflow for tumor-specific variant identification and tumor-informed MRD detection."

7/ As part of the EMBO Publications transparent editorial process initiative (see our Editorial at <http://embomolmed.embopress.org/content/2/9/329>), EMBO Molecular Medicine will publish online a Review Process File (RPF) to accompany accepted manuscripts.

This file will be published in conjunction with your paper and will include the anonymous referee reports, your point-by-point response and all pertinent correspondence relating to the manuscript. Let us know whether you agree with the publication of the RPF and as here, if you want to remove or not any figures from it prior to publication.

I look forward to receiving your revised manuscript.

Yours sincerely,

Lise Roth

***** Reviewer's comments *****

Referee #1 (Remarks for Author):

The authors have addressed my original concerns and comments. I would just caution the authors to proof-read for grammatical errors in order to improve readability.

Examples, but not limited to just these:

1. Methods, Library preparation and sequencing, page 25 line 565: "...which is effective in circularize DNA molecules..."
2. Page 31, line 695: should perhaps read "variant counts" rather than "counts of variant"
3. Page 31, line 698: "The fraction of each variant type for each patient were..."

Referee #2 (Remarks for Author):

The authors have adequately addressed the issues raised.

Referee #3 (Remarks for Author):

The authors have addressed my concerns and comments adequately

******* Reviewer's comments *********Referee #1** (Remarks for Author):

The authors have addressed my original concerns and comments. I would just caution the authors to proof-read for grammatical errors in order to improve readability.

Examples, but not limited to just these:

1. Methods, Library preparation and sequencing, page 25 line 565: "...which is effective in circularize DNA molecules..."
2. Page 31, line 695: should perhaps read "variant counts" rather than "counts of variant"
3. Page 31, line 698: "The fraction of each variant type for each patient were..."

We apologize for the grammatical errors of our manuscript. We have carefully proof-read the manuscript and corrected the grammatical errors and typos.

Referee #2 (Remarks for Author):

The authors have adequately addressed the issues raised.

Thank you for the valuable comments and suggestions on our manuscript.

Referee #3 (Remarks for Author):

The authors have addressed my concerns and comments adequately

Thank you for your professional review and constructive comments on our manuscript.

We thank all the editors and reviewers for their valuable and constructive comments on our manuscript.

Yours Sincerely,

Li Weng (on behalf of all authors)

Department of Research and Development, AccuraGen Inc

16th Jul 2024

Dear Dr. Weng,

Thank you for submitting the revised files. I have updated the Data Availability Section according to the indications provided. I am pleased to inform you that your manuscript is accepted for publication and is now being sent to our publisher to be included in the next available issue of EMBO Molecular Medicine.

If you have any questions, please do not hesitate to contact the Editorial Office.

Thank you for your contribution to EMBO Molecular Medicine, and congratulations on your interesting work!

With kind regards,

Lise Roth
